# SPOTTING EXPRESSIVITY BOTTLENECKS AND FIXING THEM OPTIMALLY

## ABSTRACT

Machine learning tasks are generally formulated as optimization problems, where one searches for an optimal function within a certain functional space. In practice, parameterized functional spaces are considered, in order to be able to perform gradient descent. Typically, a neural network architecture is chosen and fixed, and its parameters (connection weights) are optimized, yielding an architecture-dependent result. This way of proceeding however forces the evolution of the function during training to lie within the realm of what is expressible with the chosen architecture, and prevents any optimization across possible architectures. Costly architectural hyper-parameter optimization is often performed to compensate for this. Instead, we propose to adapt the architecture on the fly during training. We show that the information about desirable architectural changes, due to expressivity bottlenecks when attempting to follow the functional gradient, can be extracted from the backpropagation. To do this, we propose a new mathematically well-grounded method to detect expressivity bottlenecks on the fly and solve them by adding suitable neurons when and where needed. Thus, while the standard approach requires large networks, in terms of number of neurons per layer, for expressivity and optimization reasons, we are able to start with very small neural networks and let them grow appropriately. As a proof of concept, we show results on the MNIST dataset, matching large neural network accuracy, with competitive training time, while removing the need for standard architectural hyper-parameter search.

## 1 INTRODUCTION

**Issues with the fixed-architecture paradigm.** Universal approximation theorems such as Hornik et al. (1989) are historically among the first theoretical results obtained on neural networks, stating the family of neural networks with arbitrary width as a good candidate for a parameterized space of functions to be used in machine learning. However the current common practice in neural network training consists in choosing a fixed architecture, and training it, without any possible architecture modification meanwhile. This inconveniently prevents the direct application of these universal approximation theorems, as expressivity bottlenecks that might arise in a given layer during training will not be able to be fixed. There are two approaches to circumvent this in daily practice. Either one chooses a (very) large width, to be sure to avoid expressivity issues (Hanin & Rolnick, 2019b; Raghu et al., 2017), but then consumes extra computational power to train such big models, and often needs to reduce the model afterwards, possibly using probabilistic edges (Liu et al., 2019). Or one tries different architectures and keeps the most suitable one (in terms of performance-size compromise for instance), which multiplies the computational power by the number of trials. This latter approach relates to the Auto-DeepLearning field, where different exploration strategies over the space of architecture hyper-parameters (among other ones) have been tested, including reinforcement learning (Baker et al., 2017; Zoph & Le, 2016), Bayesian optimization techniques (Mendoza et al., 2016), and evolutionary approaches (Miller et al., 1989) (Miikkulainen et al., 2017), that rely on random tries and consequently take time for exploration. Within that line, Net2Net (Chen et al., 2015), AdaptNet (Yang et al., 2018) and MorphNet (Gordon et al., 2018) propose different strategies to explore possible variations of a given architecture, possibly guided by model size constraints. Instead, we aim at providing a way to locate precisely expressivity bottlenecks in a trained network, which might speed up neural architecture search significantly. Moreover, based on such observations,

we aim at modifying the architecture *on the fly* during training, in a single run (no re-training), using first-order derivatives only, while avoiding neuron redundancy.

**Neural architecture growth.**   A related line of work consists in growing networks neuron by neuron, by iteratively estimating the best possible neurons to add, according to a certain criterion. For instance, Wu et al. (2019) and Firefly (Wu et al., 2020) aim at escaping local minima by adding neurons that minimize the loss under neighborhood constraints. These neurons are found by gradient descent or by solving quadratic problems involving second-order derivatives. Another example is GradMax (Evci et al., 2022), which seeks to minimize the loss as fast as possible and involves another quadratic problem. However the neurons added by these approaches are possibly redundant with existing neurons, in particular if one does not wait for training convergence to a local minimum (which is time consuming) before adding neurons, therefore producing larger-than-needed architectures. On the opposite we will explicitly take redundancy into account in our growing criterion.

**Optimization properties.**   An important reason for common practice to choose wide architectures is the associated optimization properties: sufficiently larger networks are proved theoretically and shown empirically to be better optimizers than small ones Jacot et al. (2018). Typical, small networks exhibit issues with spurious local minima, while wide ones usually find good nearly-global minima. One of our goals is to train small networks without suffering from such optimization difficulties.

**Notions of expressivity.**   Several concepts of expressivity or complexity exist in the Machine Learning literature, ranging from Vapnik-Chervonenkis dimension and Rademacher complexity to the number of pieces in a piecewise affine function (as networks with ReLU activations are) Serra et al. (2018); Hanin & Rolnick (2019a). Bottlenecks have been also studied from the point of view of Information Theory, through mutual information between the activities of different layers (Tishby & Zaslavsky, 2015); this quantity is difficult to estimate though. Also relevant and from Information Theory, the Minimum Description Length paradigm and Kolmogorov complexity enable to search for a compromise between performance and model complexity.

In this article, we aim at measuring lacks of expressivity as the difference between what the back-propagation asks for and what can be done by a small parameter update (such as a gradient step), that is, between the desired variation for each activation in each layer (for each sample) and the best one that can be realized by a parameter update. Intuitively, differences arise when a layer does not have sufficient expressive power to realize the desired variation. Our main contributions are that we:

- take a functional analysis viewpoint over gradient descent on neural networks, suggesting to attempt to follow the functional gradient. We optimize not only the weights of the current architecture, but also the architecture itself on the fly, in order to progressively move towards more suitable parameterized functional spaces.
- properly define and quantify the notion of expressivity bottlenecks, globally at the neural network output as well as at each layer, and this in an easily computionable way. This allows to localize the expressivity bottlenecks, by spotting layers with great lacks of expressivity;
- mathematically define the best possible neurons to add to a given layer to decrease lacks of expressivity as a quadratic problem; compute them and their associated expressivity gain;
- check that adding these best neurons is better indeed than adding random ones;
- are able to train a neural network without gradient descent (yet still relying on backpropagation) by just adding such best neurons, without parameter update;
- naturally obtain a series of compromises between performance and number of neurons, in a single run, thus removing the need for layer width hyper-optimization, and this in competitive computational complexity with respect to classically training a large model just once. One could define a target accuracy and stop adding neurons when it is reached.

## 2   NOTATIONS AND DEFINITIONS

### 2.1   NOTATIONS

We consider a feedforward neural network with $L$ hidden layers, $f_\theta : \mathbb{R}^p \to \mathbb{R}^d$, where the parameters

$\theta := (\boldsymbol{W}_1, ..., \boldsymbol{W}_L)$ are organized into affine layers followed by activation functions $\sigma_l$. We denote the dataset by $\mathcal{D} := \{(\boldsymbol{x}_1, \boldsymbol{y}_1), ..., (\boldsymbol{x}_N, \boldsymbol{y}_N)\}$, with $\mathrm{x}_i \in \mathbb{R}^p$ and $\mathrm{y}_i \in \mathbb{R}^d$, and the loss function by $\mathcal{L}$. We will assume that $\sigma_l$ is **differentiable at 0** and that $\mathcal{L}$ is differentiable on $\mathbb{R}^d$ and that $\sigma_l(0) = 0$. Except in Part 3.4, the dataset $\mathcal{D}$ is fixed. The network iteratively computes:

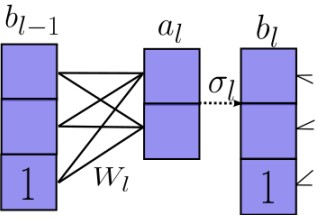

$$\boldsymbol{b}_0(\boldsymbol{x}) = \begin{pmatrix} \boldsymbol{x} \\ 1 \end{pmatrix} \qquad \boldsymbol{a}_l(\boldsymbol{x}) = \boldsymbol{W}_l \boldsymbol{b}_{l-1}(\boldsymbol{x}) \qquad \boldsymbol{b}_l(\boldsymbol{x}) = \begin{pmatrix} \sigma_l(\boldsymbol{a}_l(\boldsymbol{x})) \\ 1 \end{pmatrix}$$

with $f_\theta(\boldsymbol{x}) = \sigma_L(\boldsymbol{a}_L(\boldsymbol{x}))$. To any vector-valued function noted $\boldsymbol{t}(\boldsymbol{x})$ and any batch of inputs $\boldsymbol{X} := [\boldsymbol{x}_1, ..., \boldsymbol{x}_n]$, we associate the concatenated matrix $\boldsymbol{T}(\boldsymbol{X}) := (\boldsymbol{t}(\boldsymbol{x}_1) \quad ... \quad \boldsymbol{t}(\boldsymbol{x}_n))$. The matrices of pre-activation and post-activation activities at layer $l$ over a mini-batch $\boldsymbol{X}$ are thus respectively $\boldsymbol{A}_l(\boldsymbol{X}) = (\boldsymbol{a}_l(\boldsymbol{x}_1) \quad ... \quad \boldsymbol{a}_l(\boldsymbol{x}_n))$ and $\boldsymbol{B}_l(\boldsymbol{X}) = (\boldsymbol{b}_l(\boldsymbol{x}_1) \quad ... \quad \boldsymbol{b}_l(\boldsymbol{x}_n))$.

Figure 1: Notations

## 2.2 APPROACH

We take a functional perspective on the use of neural networks. Ideally in a machine learning task, one would search for a function $f : \mathbb{R}^p \to \mathbb{R}^d$ that minimizes the loss $\mathcal{L}$ by gradient descent: $\frac{\partial f}{\partial t} = -\nabla_f \mathcal{L}(f)$ for some metric on the functional space $\mathcal{F}$ (typically, $L_2$), where $\nabla_f$ denotes the functional gradient. For a chosen $\eta > 0$, the descent direction $\boldsymbol{v}_{\text{goal}} := -\eta \nabla_f \mathcal{L}(f)$ is a function of the same type as $f$, indicating the best infinitesimal variation to add to $f$ to decrease the loss.

In practice, to compute the gradient, a finite-dimensional parametric space of functions is considered by choosing a particular neural network architecture $\mathcal{A}$ with weights $\theta \in \Theta_{\mathcal{A}}$. The associated parametric search space $\mathcal{F}_{\mathcal{A}}$ then consists of all possible functions $f_\theta$ that can represented with such a network for any parameter value $\theta$.

**Gradient descent reminder.** For the sake of simplicity, let us consider a loss of the form $\mathcal{L}(f) = \mathbb{E}_{(\boldsymbol{x}, \boldsymbol{y}) \sim \mathcal{D}} \left[ \mathcal{L}(f(\boldsymbol{x}), \boldsymbol{y}) \right]$. Under standard weak assumptions (A.1), and up to a multiplicative learning rate, the gradient descent is then of the form:

$$\frac{\partial \theta}{\partial t} = -\nabla_\theta \mathcal{L}(f_\theta) = - \mathbb{E}_{(\boldsymbol{x}, \boldsymbol{y}) \sim \mathcal{D}} \left[ \nabla_\theta \mathcal{L}(f_\theta(\boldsymbol{x}), \boldsymbol{y}) \right]$$

Using the chain rule, this yields a function change :

$$\boldsymbol{v}_{\text{GD}} := \eta \frac{\partial f_\theta}{\partial t} = \eta \frac{\partial f_\theta}{\partial \theta} \frac{\partial \theta}{\partial t} = \frac{\partial f_\theta}{\partial \theta} \mathbb{E}_{(\boldsymbol{x}, \boldsymbol{y}) \sim \mathcal{D}} \left[ \frac{\partial f_\theta}{\partial \theta}^T (\boldsymbol{x}) \, \boldsymbol{v}_{\text{goal}}(\boldsymbol{x}) \right]$$

**Optimal move.** We name $\mathcal{T}_{\mathcal{A}}^{f_\theta}$, or just $\mathcal{T}_{\mathcal{A}}$, the tangent space of $\mathcal{F}_{\mathcal{A}}$ at $f_\theta$, that is, the set of all possible infinitesimal variations around $f_\theta$ under small parameter variations: $\mathcal{T}_{\mathcal{A}}^{f_\theta} := \{ \frac{\partial f_\theta}{\partial \theta} \, \delta\theta \mid \text{s.t. } \delta\theta \in \Theta \}$. This linear space is a first order approximation of the neighborhood of $f_\theta$ within $\mathcal{F}_{\mathcal{A}}$.

The direction $\boldsymbol{v}_{\text{GD}}$ obtained above by gradient descent is actually not the best one to consider within $\mathcal{T}_{\mathcal{A}}$. Indeed, the best move $\boldsymbol{v}^*$ would be the orthogonal projection of the desired direction $\boldsymbol{v}_{\text{goal}} := -\eta \nabla_{f_\theta} \mathcal{L}(f_\theta)$ onto $\mathcal{T}_{\mathcal{A}}$. This projection depends on the chosen metric and is what a (generalization of the notion of) natural gradient computes (Ollivier, 2017).

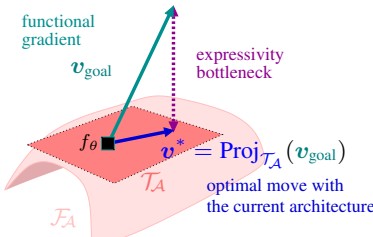

Figure 2: The expressivity bottleneck is measured as the difference between the optimal functional move $\boldsymbol{v}^*$ given the architecture $\mathcal{A}$ and the functional gradient $\boldsymbol{v}_{\text{goal}}$. The former is the projection of the latter onto the tangent space $\mathcal{T}_{\mathcal{A}}$.

**Lack of expressivity.** When $-\eta \nabla_{f_\theta} \mathcal{L}(f_\theta)$ does not belong to the reachable subspace $\mathcal{T}_\mathcal{A}$, there is a lack of expressivity, that is, the parametric space $\mathcal{A}$ is not rich enough to follow the ideal functional gradient descent. This happens frequently with small neural networks.

**Example.** Suppose one tries to estimate the function $y = f_{\text{true}}(x) = 2\sin(x) + x$ with a linear model $f_{\text{predict}}(x) = ax + b$. Consider $(a, b) = (1, 0)$ and the square loss $\mathcal{L}$. For the dataset of inputs $(x_0, x_1, x_2, x_3) = (0, 1.5, \pi, 4.5)$, there exists no parameter update $(\delta a, \delta b)$ that would improve prediction at $x_0, x_1, x_2$ and $x_3$ simultaneously, as the space of linear functions $\{f : x \to ax + b \mid a, b \in \mathbb{R}\}$ is not expressive enough. To improve the prediction at $x_0, x_1, x_2$ **and** $x_3$, one should look for another, more expressive functional space such that for $i = 0, 1, 2, 3$ the functional update $\Delta f_{\text{predict}}(x_i) := f_{\text{predict}}^{t+1}(x_i) - f_{\text{predict}}^t(x_i)$ goes into the same direction as minus the functional gradient $\boldsymbol{v}_{\text{goal}}(x_i) := -\eta \nabla_{f_{\text{predict}}(x_i)} \mathcal{L}(f_{\text{predict}}(x_i), y_i) = -\eta 2(f_{\text{predict}}(x_i) - y_i)$ where $y_i = f_{\text{true}}(x_i)$.

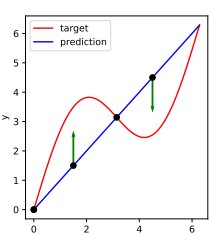

Figure 3: Linear interpolation

## 2.3 UPDATING LAYER ACTIVITIES

**Ideal updates.** The same reasoning can be applied to the pre-activations $\boldsymbol{a}_l$, seen as functions defined over the input space of the neural network. The optimal update of the weights of the different layers is then the projection of the desired update direction of pre-activation functions, i.e. $\eta \nabla_{\boldsymbol{a}_l} \mathcal{L}(f_\theta)$, obtained by back-propagation, onto the linear subspace $\mathcal{T}_\mathcal{A}^{\boldsymbol{a}_l}$ of possible variations within the architecture, as we will detail now just like in the intuition part. Training a neural network is usually done by gradient descent, which consists in updating the weight matrices $\boldsymbol{W}_l$ recursively using back-propagation. More precisely, the gradient of the loss w.r.t. $W_{i,j}^l$ is the product of the backpropagation from the loss $\mathcal{L}$ till layer preactivations $\boldsymbol{a}_l(\boldsymbol{x})$ and of the layer-specific derivative from these preactivations till the weight in question: $\frac{\partial \boldsymbol{a}_l(\boldsymbol{x})}{\partial W_{i,j}^l} \nabla_{\boldsymbol{u}} \mathcal{L}(\sigma_L(\boldsymbol{W}_L(...\sigma_l(\boldsymbol{u}))))_{|\boldsymbol{u} = \boldsymbol{a}_l(\boldsymbol{x})}$. The opposite of the second term times $\eta$ the positive real number, $\boldsymbol{v}_{\text{goal}}^l(\boldsymbol{x}) := -\eta \nabla_u \mathcal{L}(\sigma_L(\boldsymbol{W}^L(...\sigma_l(\boldsymbol{u}))))_{|\boldsymbol{u} = \boldsymbol{a}_l(\boldsymbol{x})}$, indicates the **desired update direction** for $\boldsymbol{u} = \boldsymbol{a}_l(\boldsymbol{x})$. Mathematically speaking, if at time $t$ any activity update were possible at each layer $l$, we would choose at $t + 1$ the pre-activation function updates such that for all samples $i = 1, ..., n$:

$$\Delta \boldsymbol{a}_l(\boldsymbol{x}_i) := \boldsymbol{a}_l^{t+1}(\boldsymbol{x}_i) - \boldsymbol{a}_l^t(\boldsymbol{x}_i) = -\eta \nabla_{\boldsymbol{a}_l^t(\boldsymbol{x})} \mathcal{L}(\sigma_L(\boldsymbol{W}_L(...(\boldsymbol{a}_l^t(\boldsymbol{x}))), \boldsymbol{y})_{|(\boldsymbol{x}, \boldsymbol{y}) = (\boldsymbol{x}_i, \boldsymbol{y}_i)} \quad (1)$$

Unfortunately, most of the time no parameter move $\delta\theta$ is able to induce this progression for each $\boldsymbol{x}_i$ at the same time, because the $\theta$-parameterized family of functions $\boldsymbol{a}_l$ is not expressive enough. Intuitively the ideal update above does not result from a parameter update but from an update of the pre-activation functions in their functional spaces. Following this ideal update rather than the classical gradient descent would optimally decrease the loss, with an order-1 effect in $\eta$ provided the gradient is non-0. Unlike this ideal update, the classical gradient descent decreases the global loss at first order in $\eta$ but does not necessarily improve the prediction for all $\boldsymbol{x}_i$.

For the rest of the paper we will note $\boldsymbol{v}_{\text{goal}}^l(\boldsymbol{x}_i) := -\eta \nabla_{\boldsymbol{a}_l(\boldsymbol{x})} \mathcal{L}(f_\theta(\boldsymbol{x}), \boldsymbol{y})_{|(\boldsymbol{x}, \boldsymbol{y}) = (\boldsymbol{x}_i, \boldsymbol{y}_i)}$.

**Activity update resulting from a parameter change.** Given a subset of parameters $\tilde{\theta}$, an incremental direction $\delta\tilde{\theta}$ to update $\tilde{\theta}$, and an amplitude $\eta > 0$, the impact of the parameter update $\delta\tilde{\theta}$ on the pre-activity $\boldsymbol{a}_l$ at layer $l$ at order 1 in $\delta\tilde{\theta}$ is $\boldsymbol{v}^l(\boldsymbol{x}_i, \delta\tilde{\theta}) := \frac{\partial \boldsymbol{a}_l(\boldsymbol{x})}{\partial \tilde{\theta}} \delta\tilde{\theta}$.

**Remark 1.** *We could have choose to study the desired update for $\boldsymbol{b}_l$, our choice for $\boldsymbol{a}_l$ is explained in section A.2.*

## 3 EXPRESSIVITY BOTTLENECKS

We now define expressivity bottlenecks based on the activity updates, both actual $\boldsymbol{v}^l(.)$ and desired $\boldsymbol{v}_{\text{goal}}^l(.)$ ones (cf Figure 2):

**Definition 3.1** (Lack of expressivity). *For a neural network $f_\theta$ and a minibatch of points $\{(\boldsymbol{x}_i, \boldsymbol{y}_i)\}_{i=1}^n$, we define the lack of expressivity at layer $l$ as how far the desired activity update $\boldsymbol{V}_{goal}^l$ is from the closest possible activity update realizable by a parameter change $\delta\theta$:*

$$\min_{\boldsymbol{v}^l \in \mathcal{T}_{\mathcal{A}}^{\boldsymbol{a}_l}} \sum_{i=1}^n \left\| \boldsymbol{v}^l(\boldsymbol{x}_i) - \boldsymbol{v}_{goal}^l(\boldsymbol{x}_i) \right\|^2 = \min_{\delta\theta} \left\| \boldsymbol{V}^l(\boldsymbol{X}, \delta\theta) - \boldsymbol{V}_{goal}^l(\boldsymbol{X}) \right\|_{\mathrm{Tr}}^2 \qquad (2)$$

*where $||.||$ stands for the $L_2$ norm and $||.||_{\mathrm{Tr}}$ for the Frobenius norm.*

In the two following parts we fix a minibatch $\{(\boldsymbol{x}_i, \boldsymbol{y}_i)\}_{i=1}^n$, i.e. a subset of the full dataset $\mathcal{D}$. As $\boldsymbol{X} := [\boldsymbol{x}_1, ..., \boldsymbol{x}_1]$ is then fixed, we simplify the notation: $\boldsymbol{A}_l := \boldsymbol{A}_l(\boldsymbol{X})$, $\boldsymbol{B}_l := \boldsymbol{B}_l(\boldsymbol{X})$, $\boldsymbol{V}^l := \boldsymbol{V}^l(\boldsymbol{X}) \dots$

## 3.1 BEST MOVE WITHOUT MODIFYING THE ARCHITECTURE OF THE NETWORK

Let $\delta\boldsymbol{W}_l^*$ be the solution of 2 when the parameter variation $\delta\theta$ is restricted to involve only layer $l$ parameters, i.e. $\boldsymbol{W}_l$. This move is sub-optimal in that it does not result from an update of all architecture parameters but only of the current layer ones:

$$\delta\boldsymbol{W}_l^* = \operatorname*{arg\,min}_{\delta\mathbf{W}_l \in \mathcal{M}(\boldsymbol{W}_l.\text{shape})} \left\| \boldsymbol{V}^l(\delta\mathbf{W}_l) - \boldsymbol{V}_{\text{goal}}^l \right\|_{\mathrm{Tr}}^2 \qquad (3)$$

where $\mathcal{M}(k, l)$ is the set of matrices of size $k \times l$, i.e. here of the size of $W_l$. We denote by $\boldsymbol{V}_0^{l*}$ the associated activity variation:

$$\boldsymbol{V}_0^{l*} = \delta\boldsymbol{W}_l^* \boldsymbol{B}_{l-1} \qquad\qquad \boldsymbol{v}_0^{l*}(\boldsymbol{x}_i) = \delta\boldsymbol{W}_l^* \, \boldsymbol{b}_{l-1}(\boldsymbol{x}_i)$$

**Proposition 3.1.** *The solution of 3 and its associated activity variation are:*

$$\delta\boldsymbol{W}_l^* = \frac{1}{n} \boldsymbol{V}_{goal}^l \boldsymbol{B}_{l-1}^T (\frac{1}{n} \boldsymbol{B}_{l-1} \boldsymbol{B}_{l-1}^T)^+$$

*where $P^+$ denotes the generalized inverse of matrix $P$.*

This update $\delta\boldsymbol{W}_l^*$ is not equivalent to the usual gradient descent update, whose form is $\delta\boldsymbol{W}_l^{\text{GD}} \propto \boldsymbol{V}_{\text{goal}}^l \boldsymbol{B}_{l-1}^T$. In fact $\boldsymbol{V}_0^{l*}$ is the projection of $\boldsymbol{V}_{\text{goal}}^l$ on the post-activation matrix of layer $l-1$, that is to say onto the span of all possible directions from post-activation, through the projector $\frac{1}{n} \boldsymbol{B}_{l-1}^T (\frac{1}{n} \boldsymbol{B}_{l-1} \boldsymbol{B}_{l-1}^T)^+ \boldsymbol{B}_{l-1}$. To increase expressivity if needed, we will aim at increasing this span with the most useful directions to close the gap between this best update and the desired one. Note that the update $\delta\boldsymbol{W}_l^*$ consists of a standard gradient ($\boldsymbol{V}_{\text{goal}}^l \boldsymbol{B}_{l-1}^T$) and of a (kind of) natural gradient only for the last part (projector).

## 3.2 REDUCING EXPRESSIVITY BOTTLENECK BY MODIFYING THE ARCHITECTURE

To get as close as possible to $\boldsymbol{V}_{\text{goal}}^{l+1}$ and to increase the expressive power of the current neural network, we modify each layer of its structure. At layer $l$, we add $K$ neurons $n_1, ..., n_K$ with input weights $\boldsymbol{\alpha}_1, ..., \boldsymbol{\alpha}_k$ and output weights $\boldsymbol{\omega}_1, ..., \boldsymbol{\omega}_K$ (cf Figure 4). We have the following change : $\boldsymbol{W}_l^T \leftarrow \begin{pmatrix} \boldsymbol{W}_l^T & \boldsymbol{\alpha}_1 & ... & \boldsymbol{\alpha}_K \end{pmatrix}$ and $\boldsymbol{W}_{l+1} \leftarrow \begin{pmatrix} \boldsymbol{W}_{l+1} & \boldsymbol{\omega}_1 & ... & \boldsymbol{\omega}_K \end{pmatrix}$.
We note this modification of architecture $\theta \leftarrow \theta \oplus \theta_{l\leftrightarrow l+1}^K$ where $\oplus$ is the concatenation sign and $\theta_{l\leftrightarrow l+1}^K := (\boldsymbol{\alpha}_k, \boldsymbol{\omega}_k)_{k=1}^K$ are the added neurons.

The added neurons could be chosen randomly, as in usual neural network initialisation, but this would not yield any guarantee regarding the impact on the system loss. Another possibility would be to set either input weights $(\boldsymbol{\alpha}_k)_{k=1}^K$ or output weights $(\boldsymbol{\omega}_k)_{k=1}^K$ to 0, so that the function $f_\theta(.)$ would not be modified, while its gradient w.r.t. $\theta$ would be enriched from the new parameters. Another option is to solve a optimization problem as in the previous section with the modified structure $\theta \leftarrow \theta \oplus \theta_{l\leftrightarrow l+1}^K$ and jointly search for both the optimal new parameters $\theta_{l\leftrightarrow l+1}^K$ and the optimal variation $\delta\mathbf{W}_l$ of the old ones.

$$\operatorname*{arg\,min}_{\theta_{l\leftrightarrow l+1}^K, \delta\mathbf{W}_l} \left\| \boldsymbol{V}_{\text{goal}}^{l+1} - \boldsymbol{V}^{l+1}((\delta\mathbf{W}_l, \theta_{l\leftrightarrow l+1}^K)) \right\|_{\mathrm{Tr}}^2$$

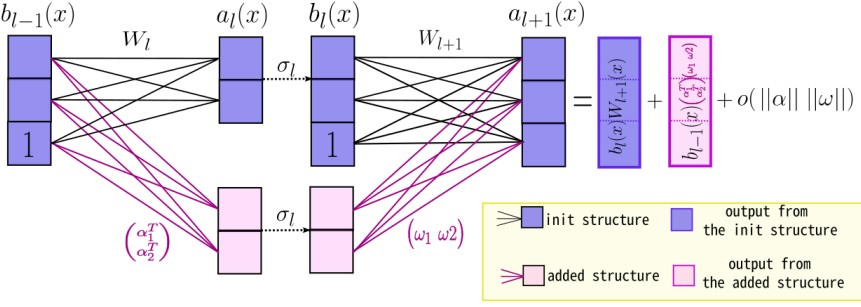

Figure 4: Adding two neurons at layer $l$ in purple ($K = 2$), with connections in purple. We have $\boldsymbol{\alpha}_i \in \mathrm{R}^3$ et $\boldsymbol{\omega}_i \in \mathrm{R}^3$ for $i = 1, 2$.

As the displacement $\boldsymbol{V}^{l+1}$ at layer $l + 1$ is actually a sum of the moves induced by the neurons already present ($\delta\mathbf{W}_l$) and by the added neurons ($\theta^K_{l\leftrightarrow l+1}$), our problem rewrites as :

$$\underset{\theta^K_{l\leftrightarrow l+1}, \delta\mathbf{W}_l}{\arg\min} ||\boldsymbol{V}_{\text{goal}}^{l+1} - \boldsymbol{V}^{l+1}(\delta\mathbf{W}_l) - \boldsymbol{V}^{l+1}(\theta^K_{l\leftrightarrow l+1})||^2_{\text{Tr}} \tag{4}$$

With $\boldsymbol{v}^{l+1}(\boldsymbol{x}, \theta^K_{l\leftrightarrow l+1}) := \sum_{j=1}^K \boldsymbol{w}_k \, (b_{l-1}(\boldsymbol{x})^T \boldsymbol{\alpha}_k)$. Refering to the definition of $\boldsymbol{v}^{l+1}(\boldsymbol{x})$ this choice have to be explained because the partial derivative with respect to $(\boldsymbol{\alpha}_k, \boldsymbol{\omega}_k)$ is actually 0 (See A.2). We solve this problem in two steps. Let us fix for the moment $\delta\mathbf{W}_l \in \mathcal{M}(|\boldsymbol{v}_{\text{goal}}^{l+1}(\boldsymbol{x})|, |\boldsymbol{b}_l(\boldsymbol{x})|)$, standing for an update of the matrix $\boldsymbol{W}_{l+1}$, and search for the best new parameters $\theta^K_{l\leftrightarrow l+1}$. We note $\boldsymbol{V}_{\text{goal}_{proj}}^{l+1} = \boldsymbol{V}_{\text{goal}_{proj}}^{l+1}(\delta\mathbf{W}_l) := \boldsymbol{V}_{\text{goal}}^{l+1} - \boldsymbol{V}^{l+1}(\delta\mathbf{W}_l)$. We are looking for the following quantity :

$$\Big( \overbrace{(\hat{\boldsymbol{\alpha}}_k^*, \hat{\boldsymbol{\omega}}_k^*)_{k=1}^{K^*}}^{\hat{\theta}^{K,*}_{l\leftrightarrow l+1}}, K^* \Big) := \underset{(\boldsymbol{\alpha}_k, \boldsymbol{\omega}_k)_{k=1}^K, K}{\arg\min} \left\{ ||\boldsymbol{V}_{\text{goal}_{proj}}^{l+1} - \boldsymbol{V}^{l+1}(\theta^K_{l\leftrightarrow l+1})||^2_{\text{Tr}} \right\} \tag{5}$$

We define the matrices $\boldsymbol{N} := \frac{1}{n}\boldsymbol{B}_{l-1}\big(\boldsymbol{V}_{\text{goal}_{proj}}^{l+1}\big)^T$ and $\boldsymbol{S} := \frac{1}{n}\boldsymbol{B}_{l-1}\boldsymbol{B}_{l-1}^T$. Note that $\boldsymbol{N}$ depends on $\delta\mathbf{W}_l$. Using the low-rank matrix approximation theorem (Eckart & Young, 1936), we can solve this quadratic optimization problem as follows.

As $\boldsymbol{S}$ is semi-positive definite, let us denote its Cholesky decomposition by $\boldsymbol{S} = \boldsymbol{S}^{1/2}\boldsymbol{S}^{1/2^T}$, and consider the SVD of the matrix $\boldsymbol{S}^{1/2^{-1}}\boldsymbol{N} = \sum_{k=1}^R \lambda_k \boldsymbol{u}_k \boldsymbol{v}_k^T$ with $\lambda_1 \geq ... \geq \lambda_R \geq 0$, where $R$ is the rank of the matrix $\boldsymbol{N}$. Then:

**Proposition 3.2.** *The solution of (5) can be written as:*

- *optimal number of neurons: $K^* = R$*

- *their optimal weights: $(\hat{\boldsymbol{\alpha}}_k^*, \hat{\boldsymbol{\omega}}_k^*) = (\boldsymbol{S}^{1/2^{T^{-1}}}\boldsymbol{u}_k, \boldsymbol{v}_k)$ for $k = 1, ..., R$.*

*Moreover for any number of neurons $K \leqslant R$, and associated scaled weights $\hat{\theta}^{K,*}_{l\leftrightarrow l+1}$, the expressivity gain and the first order in $\eta$ of the loss improvement due to the addition of these $K$ neurons are equal and can be quantified very simply as a function of the eigenvalues $\lambda_k$:*

$$\frac{1}{n}||\boldsymbol{V}_{goal_{proj}}^{l+1} - \boldsymbol{V}^{l+1}(\hat{\theta}^{K,*}_{l\leftrightarrow l+1})||^2_{\text{Tr}} = \frac{1}{n}||\boldsymbol{V}_{goal_{proj}}^{l+1}||^2_{\text{Tr}} - \boxed{\sum_{k=1}^K \lambda_k^2}$$

$$\frac{1}{n}\sum_{i=1}^n \mathcal{L}(f_{\theta \oplus \hat{\theta}^{K,*}_{l\leftrightarrow l+1}}(\boldsymbol{x}_i), \boldsymbol{y}_i) = \frac{1}{n}\sum_{i=1}^n \mathcal{L}(f_\theta(\boldsymbol{x}_i), \boldsymbol{y}_i) + \frac{\sigma_l'(0)}{\eta}\boxed{\sum_{k=1}^K \lambda_k^2} + o(||\hat{\theta}^{K,*}_{l\leftrightarrow l+1}||^2)$$

**Proposition 3.3.** *If $S$ is positive definite, then solving (5) is equivalent to taking $\omega_k = N\alpha_k$ and finding the $K$ first eigenvectors $\alpha_k$ associated to the $K$ largest eigenvalues $\lambda$ of the generalized eigenvalue problem :*

$$NN^T\alpha_k = \lambda S\alpha_k$$

This formulation is useful when dimensions of $N$ and $S$ are large. Considering LOBPCG method (Peter Benner, 29 Apr 2020) allows not to invert the matrix and to compute the Cholesky factorization in proposition **3.2**. In practice the matrix $S$ is positive definite except for $l - 1 = 0$, and even in this last case it is possible to define the Cholesky decomposition of $S$ (cf Appendix).

**Corollary.** *The matrix $\delta\mathbf{W}_l$ which minimizes (6) (through its impact on $V_{goal\ proj}^{l+1}$) or equivalently minimizes the sum of orders zero and one in $\eta$ of (7) is given by $\delta W_{l+1}^*$ in Proposition **3.1**.*

**Corollary.** *For all integers $m, m'$ such that $m + m' \leqslant R$, at order one in $\eta$ it is equivalent to add $m + m'$ neurons simultaneously according to the previous method or to add $m$ neurons then $m'$ neurons by applying successively the previous method twice.*

Minimizing the distance (6), ie the distance between $V_{\text{goal}}^{l+1}(\delta\mathbf{W}_l)$ and $V^{l+1}(\theta_{l\leftrightarrow l+1})$, is equivalent to minimize the loss $\mathcal{L}$ at order one in $\gamma$, and it is directly due to the following development:

$$\mathcal{L}(f_{\theta\oplus\theta_{l\leftrightarrow l+1}^K}) = \mathcal{L}(f_\theta) - \sigma_l'(0)\frac{1}{\eta}\frac{1}{n}\sigma_l'(0)\Big\langle V_{\text{goal}}^{l+1}(\delta\mathbf{W}_l),\ V^{l+1}(\theta_{l\leftrightarrow l+1})\Big\rangle_{\text{Tr}} + o(||V^{l+1}(\theta_{l\leftrightarrow l+1})||)$$

When solving (6), we notice that the family $\{V^{l+1}((\alpha_k, \omega_k))\}_{k=1}^K$ of pre-activity variations induced by adding the neurons $\hat{\theta}_{l\leftrightarrow l+1}^{K,*}$ is orthogonal for the trace scalar product. We could say that the added neurons are orthogonal to each other in that sense. The addition of each neuron $k$ has an impact on the order of $\lambda_k$, which can be used to define a criterion to decide whether the neuron $k$ should be added to the layer or not, i.e. only if $\lambda_k^2 \mathcal{L}(f_\theta) > \tau$. We name the operation $\theta \leftarrow \theta \oplus \hat{\theta}_{l\leftrightarrow l+1}^{K,*}$ the ***K-update of the network at layer*** $l$.

## 3.3 CHOOSING THE AMPLITUDE FACTOR $\gamma$ WHEN ADDING NEURONS

Let us consider the $l+1$-th layer of the network. Having the best update for linear layer $l+1$, $\delta W_{l+1}^*$, and the best neurons to add accordingly to layer $l$, $(\hat{\alpha}_k^*, \hat{\omega}_k^*)_{k=1}^K$, we estimate the best factors $\gamma_0$ and $\gamma$ to multiply the new neurons with, in order to speed up the learning. Defining the updates $\theta^{\delta(l+1)}(\gamma_0) = (W_1, ..., W_{l+1} + \gamma_0\,\delta W_{l+1}^*, ..., W_L)$ and $\hat{\theta}_{l\leftrightarrow l+1}^{K,*}(\gamma) = (\gamma\hat{\alpha}_k^*, \hat{\omega}_k^*)_{k=1}^K$, we apply a line search algorithm to find a local minimum of the loss function :

$$\gamma_0^* := \underset{\gamma_0 \in \mathcal{V}(0)}{\arg\min}\ \frac{1}{n}\sum_{i=1}^n \mathcal{L}(f_{\theta^{\delta(l+1)}(\gamma_0)}(X_i), Y_i) \tag{6}$$

$$\gamma^* := \underset{\gamma \in \mathcal{V}(0)}{\arg\min}\ \frac{1}{n}\sum_{i=1}^n \mathcal{L}(f_{\theta^{\delta(l+1)}(\gamma_0^*)\oplus\hat{\theta}_{l\leftrightarrow l+1}^{K,*}(\gamma)}(X_i), Y_i) \tag{7}$$

where $\mathcal{V}(0)$ is a positive neighbourhood of zero.
*Remark : the amplitude factor can also be defined differently, for example by choosing a different amplitude factor for the input and output parameters, ie $(\sqrt{\gamma_1}\alpha_k, \sqrt{\gamma_2}\omega_k)_{k=1}^K$.*

## 3.4 VARIANCE OF THE ESTIMATOR

Noting $n_l$ the number of neurons already in layer $l$ before the *K-update*, we discuss the variance of our estimate $\delta\hat{\theta}_{l\leftrightarrow l+1}^{K,*}$ for $l = 1, ..., L - 1$, as a function of the minibatch. At layer $l + 1$, the solution of (5) is an estimator of $\theta_{l\leftrightarrow l+1}^{K,*} := (\alpha_k^*, \omega_k^*)_{k=1}^K$, which is the minimizer of :

$$\theta_{l\leftrightarrow l+1}^{K,*} := \underset{\alpha, \omega}{\arg\min}\ \mathbb{E}\left[\left\|v_{\text{goal}\ proj}^{l+1}(X_i) - \sum_{k=1}^K v_k^{l+1}(X_i)\right\|^2\right] \tag{8}$$

where the expectation is taken over all possible random samples $X_i \sim \mathcal{D}$. The variance of the directions $(\hat{\alpha}_k^*, \hat{\omega}_k^*)_{k=1}^K$ grows in $\sqrt{p}$ where $p := \max(n_{l-1}, n_{l+1})$ and decreases in $\sqrt{N}$ where $N$ is the minibatch size used for the estimation of (8). In practice we will start with $N = 100$ and increase $N$ with the architecture growth.

## 4 EXPERIMENTAL RESULTS

In Figures 5 and 6, we run an experiment on the MNIST dataset ((LeCun et al., 1998)), with 7 CPUs, repeated 20 times. All models are trained for 50 seconds with $Adam(lr = 0.0001, \mu = 0)$ with constant mini-batch of size 100. The activation functions are $\sigma_l = selu$ if $l = 1, 2$ and $\sigma_3 = Softmax$. We plot accuracy mean and standard deviation on test set for our approach with architecture growth and for standard training with a fixed architecture. For our approach, we start with a feedforward model with 2 hidden layers of widths $[1, 1]$, initialized with Kaiming normal. Every 0.05 seconds (with Adam(lr = 0.0001)), we extend the two hidden layers according to our method. We compare the performance of our method with classical training of large models (Fig. 5), or of models with the same final architecture as ours (Fig. 6), to check performance when one already knows the correct architecture. Classic models are initialized with Kaiming normal. More graphics can be found in Appendix C. We note that large models take more computational time to train, and that architecture growth yields better or similar performance while avoiding layer width tuning.

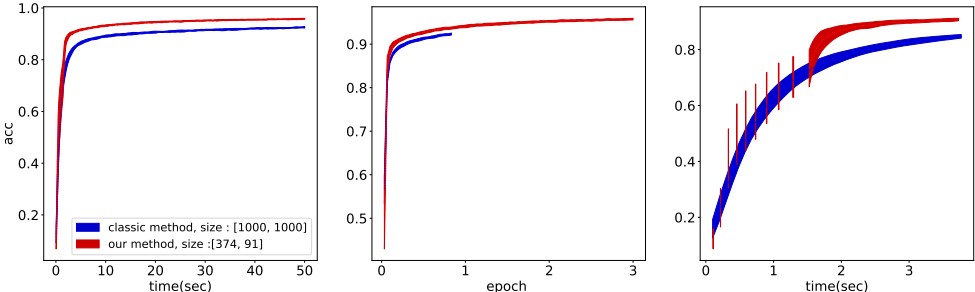

Figure 5: All graphics represent the same experiment but from a different perspective. **Left** : after partitioning computational time on intervals of size 0.1 seconds, we compute a linear interpolation for the accuracy value. **Middle** : the accuracy value against number of epochs, where the time needed to compute the optimal neurons is not noticeable. **Right** : accuracy against computational time, where durations due to Cholesky decompositions and their happening instants are averaged over experiments, for better visualisation purposes.

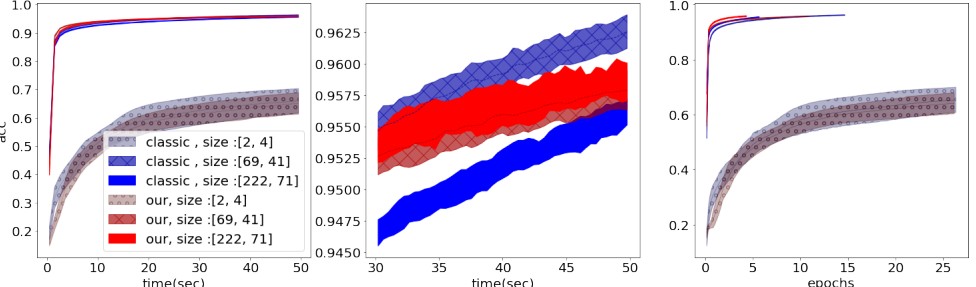

Figure 6: **Left** : we plot the interpolation of accuracy on intervals of size 0.1 second. **Middle** : zoom of the top left plot. **Right** : the accuracy value against number of epochs, unlike middle plot in graphic 5, our method is trained on fewer epochs compared to classic model, indeed with equal architecture our method spends time computing the best neurons while the classic method continues its training.

## 5 ABOUT GREEDY GROWTH SUFFICIENCY

One might wonder whether a greedy approach on layer growth might get stuck in a non-optimal state. We derive the following series of propositions in this regard. Since in this work we add neurons layer per layer independently, we study here the case of a single hidden layer network, to spot potential layer growth issues. For the sake of simplicity, we consider the task of least square regression towards an explicit continuous target $f^*$, defined on a compact set. That is, we aim at minimizing the loss:

$$\inf_f \sum_{x \in \mathcal{D}} \|f(x) - f^*(x)\|^2$$

where $f(x)$ is the output of the neural network and $\mathcal{D}$ is the training set.

**Proposition 5.1** (Greedy completion of an existing network). *If $f$ is not $f^*$ yet, then there exists a set of neurons to add to the hidden layer such that the new function $f'$ will have a lower loss than $f$.*

One can even choose the added neurons such that the loss is arbitrarily well minimized. Furthermore:

**Proposition 5.2** (Greedy completion by one single neuron). *If $f$ is not $f^*$ yet, there exists a neuron to add to the hidden layer such that the new function $f'$ will have a lower loss than $f$.*

As a consequence, there exists no situation where one would need to add many neurons simultaneously to decrease the loss: it is always feasible with a single neuron. One can express a lower bound on how much the loss has improved (for the best such neuron), but it is not a very good bound without further assumptions on $f$.

**Proposition 5.3** (Greedy completion by one infinitesimal neuron). *The neuron in the previous proposition can be chosen to have arbitrarily small input weights.*

This detail is important in that our approach is based on the tangent space of the function $f$ and consequently manipulates infinitesimal quantities. Though we perform line search in a second step and consequently add non-infinitesimal neurons, our first optimization problem relies on the linearization of the activation function by requiring the added neuron to have infinitely small input weights, to make the problem easier to solve. This proposition confirms that such neuron exists indeed.

**Correlations and higher orders.** Note that, as a matter of fact, our approach exploits linear correlations between inputs of a layer and desired output variations. It might happen that the loss is not minimized yet but there is no such correlation to exploit anymore. In that case the optimization problem (5) will not find neurons to add. Yet following Prop. 5.3 there does exist a neuron with arbitrarily small input weights that can reduce the loss. This paradox can be explained by pushing further the Taylor expansion of that neuron output in terms of weight amplitude (single factor $\varepsilon$ on all of its input weights), for instance $\sigma(\varepsilon\boldsymbol{\alpha}\cdot\boldsymbol{x}) \simeq \sigma(0) + \sigma'(0)\varepsilon\boldsymbol{\alpha}\cdot\boldsymbol{x} + \frac{1}{2}\sigma''(0)\varepsilon^2(\boldsymbol{\alpha}\cdot\boldsymbol{x})^2 + O(\varepsilon^3)$. Though the linear term $\boldsymbol{\alpha}\cdot\boldsymbol{x}$ might be uncorrelated over the dataset with desired output variation, i.e. $\mathbb{E}_{\boldsymbol{x}\sim\mathcal{D}}[\boldsymbol{\alpha}\cdot\boldsymbol{x}] = 0$, the quadratic term $(\boldsymbol{\alpha}\cdot\boldsymbol{x})^2$, or higher-order ones otherwise, might be correlated. Finding neurons with such higher-order correlations can be done by increasing accordingly the power of $(\boldsymbol{\alpha}\cdot\boldsymbol{x})$ in the optimization problem (4). Note that one could consider other function bases that the polynomials from Taylor expansion. In all cases, one does not need to solve such problems exactly but just to find an approximate solution, i.e. a neuron improving the loss.

**Adding random neurons.** Another possibility to suggest additional neurons, when expressivity bottlenecks are detected but no correlation (up to order $p$) can be exploited anymore, is to add random neurons. The first $p$ order Taylor expansions will show 0 correlation with desired output variation, hence no loss improvement nor worsening, but the correlation of the $p+1$-th order will be non-0, with probability 1, in the spirit of random projections. The loss can then be improved, all the more with a line search to optimize the neuron amplitude.

However, such random neurons also contribute to other directions in the functional space than the desired one. This hinders the loss improvement expectable from them, as the line search will need to find a compromise with the loss changes brought by these other directions. This is confirmed experimentally in Appendix C.4. To alleviate this, in the spirit of common neural network training practice, one could consider brute force combinatorics by adding many random neurons and hoping that one will be close enough to the desired direction. The difference with standard training is that we would perform such computationally-costly searches only when and where relevant, exploiting all simple information (linear correlations in each layer) first.

## 6 CONCLUSION

We have properly defined lacks of expressivity, and their minimization has allowed us to optimize the architecture on the fly, to better follow the functional gradient, enabling architecture growth. Apart from straightforward extension to other types of layers (such as convolutions) and to the addition of new layers, future work will pay attention to overfit possibilities (which we have not we have not observed so far, thanks to the optimally small number of parameters) and to neuron addition strategies.

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

# A   ASSUMPTIONS

## A.1   MEASURE THEORY STATEMENT

Let $X$ be an open subset of $\mathbf{R}$, and $\Omega$ be a measure space. Suppose $f : X \times \Omega \to \mathbf{R}$ satisfies the following conditions:

- $f(x, \omega)$ is a Lebesgue-integrable function of $\omega$ for each $x \in X$.
- For almost all $\omega \in \Omega$, the partial derivative $f_x$ or $f$ accordinf to $x$ exists for all $x \in X$.
- There is an integrable function $\theta : \Omega \to \mathbf{R}$ such that $|f_x(x, \omega)| \le \theta(\omega)$ for all $x \in X$ and almost every $\omega \in \Omega$.

Then, for all $x \in X$,

$$\frac{d}{dx} \int_\Omega f(x, \omega)\, d\omega = \int_\Omega f_x(x, \omega)\, d\omega$$

See proof and details :mea.

## A.2   REMARKS

When increasing the size of layer $l$ with $\theta_{l \leftrightarrow l+1}^K := (\boldsymbol{\alpha}_k, \boldsymbol{\omega}_k)_{k=1}^K$ starting with $(\boldsymbol{\alpha}_k, \boldsymbol{\omega}_k)_{k=1}^K = 0$, the outcome for $\boldsymbol{v}^{l+1}(\boldsymbol{x}, \theta_{l \leftrightarrow l+1}^K)$ is 0 because the gradient with respect to $(\boldsymbol{\alpha}_k, \boldsymbol{\omega}_k)_{k=1}^K$ is 0 in $(\boldsymbol{\alpha}_k, \boldsymbol{\omega}_k)_{k=1}^K$ is 0. In mathematical terms :

$$\boldsymbol{v}^{l+1}(\boldsymbol{x}, \theta_{l \leftrightarrow l+1}^K) := \frac{\partial \boldsymbol{a}_{l+1}(\boldsymbol{x})}{\partial \theta_{l \leftrightarrow l+1}^K}_{|\theta_{l \leftrightarrow l+1}^K = 0} \qquad \theta_{l \leftrightarrow l+1}^K = 0 \tag{9}$$

The impact of this modification of structure has to be seen differently. The first point of view is to say that we choose $(\boldsymbol{\omega}_k)_{k=1}^k$ then compute $\boldsymbol{v}^{l+1}(\boldsymbol{x}, (\boldsymbol{\alpha}_k)_{k=1}^K)$ as a function of the family $(\boldsymbol{\omega}_k)_{k=1}^K$. We have then :

$$\boldsymbol{v}^{l+1}(\boldsymbol{x}, \theta_{l \leftrightarrow l+1}^K) := \boldsymbol{v}^{l+1}(\boldsymbol{x}, (\boldsymbol{\alpha}_k)_{k=1}^K) = \frac{\partial \boldsymbol{a}_{l+1}(\boldsymbol{x})}{\partial \left( (\boldsymbol{\alpha}_k)_{k=1}^K \right)}_{|(\boldsymbol{\alpha}_k)_{k=1}^K = 0} (\boldsymbol{\alpha}_k)_{k=1}^K = \sum_{k=1}^K \boldsymbol{\omega}_k \boldsymbol{b}_{l-1}(\boldsymbol{x})^T \boldsymbol{\alpha}_k$$

This is equivalent to say that for each family $(\boldsymbol{\omega}_k)_{k=1}^K$, the tangent space in $\boldsymbol{a}_{l+1}$ restricted to move in the family $(\boldsymbol{\alpha}_k)_{k=1}^K$, ie $\mathcal{T}_{\mathcal{A}}^{\boldsymbol{a}_{l+1}} := \{ \frac{\partial \boldsymbol{a}_{l+1}}{\partial (\boldsymbol{\alpha}_k)_{k=1}^K}_{|(\boldsymbol{\alpha}_k)_{k=1}^K = 0} (\boldsymbol{\alpha}_k)_{k=1}^K | (\boldsymbol{\alpha}_k)_{k=1}^K \in \left( \mathbf{R}^{|\boldsymbol{b}_{l-1}(\boldsymbol{x})|} \right)^K \}$ varies with the family $(\boldsymbol{\omega}_k)_{k=1}^K$, ie $\mathcal{T}_{\mathcal{A}}^{a_{l+1}} := \mathcal{T}_{\mathcal{A}}^{a_{l+1}}((\boldsymbol{\omega}_k)_{k=1}^K)$. Optimizing over the $\boldsymbol{\omega}_k$ is equivalent to search for the better tangent space while optimizing on the $\boldsymbol{\alpha}_k$ is equivalent to find the better projection on the tangent space defined by the $\boldsymbol{\omega}_k$.

Note that making the derivative according to the $\boldsymbol{\alpha}_k$ ease the problem by removing the nonlinearity in $\sigma_l$. When reversing the roles of the $\boldsymbol{\alpha}_k$ and of the $\boldsymbol{\omega}_k$, ie fixing the $\boldsymbol{\alpha}_k$ and compute $\boldsymbol{v}^{l+1}(\boldsymbol{x}, (\boldsymbol{\omega}_k)_{k=1}^K)$, it makes the problem harder to solve because the non linearity in $\sigma_l$ remains in the optimisation problem.

Taking an other point of view, you can consider the second order of $\boldsymbol{a}_{l+1}(\boldsymbol{x})$ in $(\boldsymbol{\alpha}_k, \boldsymbol{\omega}_k)_{k=1}^K$ in 0 to recover the same expression. Indeed taking the Taylor expansion in $(\boldsymbol{\alpha}_k, \boldsymbol{\omega}_k)_{k=1}^K$

$$\boldsymbol{a}_{l+1}(\boldsymbol{x}) = \boldsymbol{a}_{l+1}(\boldsymbol{x})_{|(\boldsymbol{\alpha}_k, \boldsymbol{\omega}_k)_{k=1}^K = 0} + \sum_{k=1}^K \boldsymbol{\omega}_k \boldsymbol{b}_{l-1}(\boldsymbol{x})^T \boldsymbol{\alpha}_k + o\big( (||(\boldsymbol{\alpha}_k)_{k=1}^K|| + ||(\boldsymbol{\omega}_k)_{k=1}^K||)^2 \big)$$

[TODO] A faire

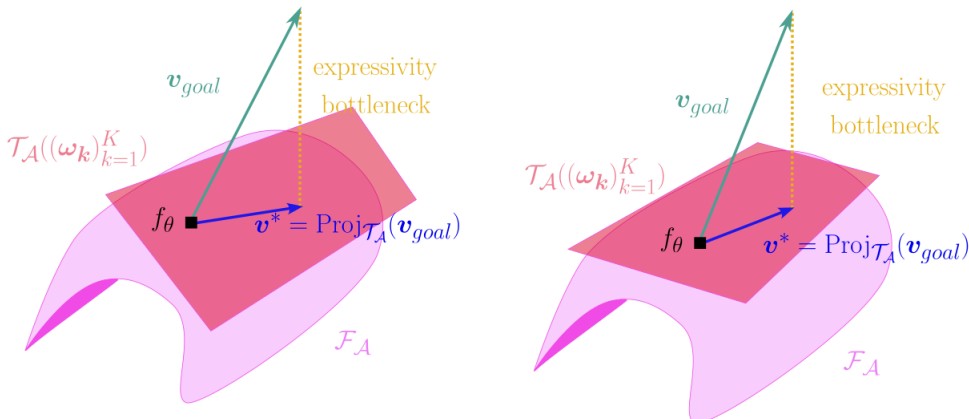

Figure 7: changing the tangent space with different values for the family $(\boldsymbol{\omega}_k)_{k=1}^K$.

# B  PROOFS

## B.1  PROPOSITION 3.1

Define $\delta \mathbf{W}_l^+$ the generalized inverse of $\delta \mathbf{W}_l$ then :

$$\delta \boldsymbol{W}_l^* = \frac{1}{n} \boldsymbol{V}_{\text{goal}}^{\,l} \boldsymbol{B}_{l-1}^T \big(\frac{1}{n} \boldsymbol{B}_{l-1} \boldsymbol{B}_{l-1}^T\big)^+ \text{ and } \boldsymbol{V}_0^l = \frac{1}{n} \boldsymbol{V}_{\text{goal}}^{\,l} \boldsymbol{B}_{l-1}^T \big(\frac{1}{n} \boldsymbol{B}_{l-1} \boldsymbol{B}_{l-1}^T\big)^+ \boldsymbol{B}_{l-1}$$

*Proof*
Consider the function $g(\delta \mathbf{W}_l) = ||\boldsymbol{V}_{\text{goal}}^{\,l} - \delta \mathbf{W}_l \boldsymbol{B}_{l-1}||_{\text{Tr}}^2$, then

$$\begin{aligned}
g(\delta \mathbf{W}_l + \boldsymbol{H}) &= ||\boldsymbol{V}_{\text{goal}}^{\,l} - \delta \mathbf{W}_l \boldsymbol{B}_{l-1} - \boldsymbol{H} \boldsymbol{B}_{l-1}||_{\text{Tr}}^2 \\
&= g(\delta \mathbf{W}_l) - 2\langle \boldsymbol{V}_{\text{goal}}^{\,l} - \delta \mathbf{W}_l \boldsymbol{B}_{l-1}, \boldsymbol{H} \boldsymbol{B}_{l-1}\rangle_{\text{Tr}} + o(||\boldsymbol{H}||) \\
&= g(\delta \mathbf{W}_l) - 2 \operatorname{Tr}\left(\big(\boldsymbol{V}_{\text{goal}}^{\,l} - \delta \mathbf{W}_l \boldsymbol{B}_{l-1}\big)^T \boldsymbol{H} \boldsymbol{B}_{l-1}\right) + o(||\boldsymbol{H}||) \\
&= g(\delta \mathbf{W}_l) - 2 \operatorname{Tr}\left(\boldsymbol{B}_{l-1}\big(\boldsymbol{V}_{\text{goal}}^{\,l} - \delta \mathbf{W}_l \boldsymbol{B}_{l-1}\big)^T \boldsymbol{H}\right) + o(||\boldsymbol{H}||) \\
&= g(\delta \mathbf{W}_l) - 2\langle \big(\boldsymbol{V}_{\text{goal}}^{\,l} - \delta \mathbf{W}_l \boldsymbol{B}_{l-1}\big) \boldsymbol{B}_{l-1}^T, \boldsymbol{H}\rangle_{\text{Tr}} + o(||\boldsymbol{H}||)
\end{aligned}$$

By identification $\nabla_{\delta \mathbf{W}_l} g(\delta \mathbf{W}_l) = 2\big(\boldsymbol{V}_{\text{goal}}^{\,l} - \delta \mathbf{W}_l \boldsymbol{B}_{l-1}\big) \boldsymbol{B}_{l-1}^T$

$$\nabla_{\delta \mathbf{W}_l} g(\delta \mathbf{W}_l) = 0 \implies \boldsymbol{V}_{\text{goal}}^{\,l} \boldsymbol{B}_{l-1}^T = \delta \mathbf{W}_l \boldsymbol{B}_{l-1} \boldsymbol{B}_{l-1}^T$$

Using the definition of the generalized inverse of $M^+$:

$$\delta \boldsymbol{W}_l^* = \frac{1}{n} \boldsymbol{V}_{\text{goal}}^{\,l} \boldsymbol{B}_{l-1}^T \big(\frac{1}{n} \boldsymbol{B}_{l-1} \boldsymbol{B}_{l-1}^T\big)^+$$

## B.2  PROPOSITION 3.2

If $\boldsymbol{S}$ is positive definite, consider the Cholesky decomposition $\boldsymbol{S} = \boldsymbol{S}^{1/2} \boldsymbol{S}^{1/2^T}$, note $R$ the rank of the matrix $\boldsymbol{N}$ and the SVD of the matrix $\boldsymbol{S}^{1/2^{-1}} \boldsymbol{N} = \sum_{k=1}^R \lambda_k \boldsymbol{u}_k \boldsymbol{v}_k^T$ with $0 \le \lambda_1 \le ... \le \lambda_R$ then the solution of (4) is written as $\hat{K}^* = R$ and $(\hat{\boldsymbol{\alpha}}_k^*, \hat{\boldsymbol{\omega}}_k^*) = (\boldsymbol{S}^{1/2^{T^{-1}}} \boldsymbol{u}_k, \boldsymbol{v}_k)$ for $k = 1, ..., R_{l+1}$. Moreover for all $K \le R$ and $\hat{\theta}_{l \leftrightarrow l+1}^{K,*} := (\hat{\boldsymbol{\alpha}}_k^*, \hat{\boldsymbol{\omega}}_k^*)_{k=1}^K$ and for $\gamma$ positive, $\hat{\theta}_{l \leftrightarrow l+1}^{K,*}(\gamma) :=$

$(\gamma\hat{\boldsymbol{\alpha}}_k^*, \hat{\boldsymbol{\omega}}_k^*)_{k=1}^K$, we have that:

$$\frac{1}{n}||\boldsymbol{V}_{\text{goal}_{proj}}^{l+1} - \boldsymbol{V}^{l+1}(\hat{\theta}_{l\leftrightarrow l+1}^K)||_{\text{Tr}}^2 = \frac{1}{n}||\boldsymbol{V}_{\text{goal}_{proj}}^{l+1}||_{\text{Tr}}^2 - \sum_{k=1}^K \lambda_k^2 \tag{10}$$

$$\frac{1}{n}\sum_{i=1}^n L(f_{\theta\oplus\hat{\theta}_{l\leftrightarrow l+1}^{K,*}(\gamma)}(\boldsymbol{x}_i), \boldsymbol{y}_i) = \frac{1}{n}\sum_{i=1}^n L(f_\theta(\boldsymbol{x}_i), \boldsymbol{y}_i) - \sigma_l'(0)\frac{\gamma}{\eta}\sum_{k=1}^K \lambda_k^2 + o(\gamma) \tag{11}$$

*Proof*

$$\underset{\hat{\theta}_{l\leftrightarrow l+1}^K}{\arg\min}\left\{\frac{1}{n}||\boldsymbol{V}_{\text{goal}_{proj}}^{l+1} - \boldsymbol{V}^{l+1}(\hat{\theta}_{l\leftrightarrow l+1}^K)||_{\text{Tr}}^2\right\} = \underset{\hat{\theta}_{l\leftrightarrow l+1}^K}{\arg\min}\left\{-\frac{2}{n}\left\langle\boldsymbol{V}_{\text{goal}_{proj}}^{l+1}, \boldsymbol{V}^{l+1}(\hat{\theta}_{l\leftrightarrow l+1}^K)\right\rangle_{\text{Tr}} + \frac{1}{n}||\boldsymbol{V}^{l+1}(\hat{\theta}_{l\leftrightarrow l+1}^K)||_{\text{Tr}}^2\right\}$$

$$= \underset{\hat{\theta}_{l\leftrightarrow l+1}^K}{\arg\min}\frac{1}{n}g(\hat{\theta}_{l\leftrightarrow l+1}^K)$$

We note

$$\frac{1}{n}g(\hat{\theta}_{l\leftrightarrow l+1}^K) = -\frac{2}{n}\sum_i^n\sum_k \boldsymbol{v}_{\text{goal}_{proj}}^{l+1}(\boldsymbol{x}_i)^T\left(\boldsymbol{\alpha}_k^T\boldsymbol{b}_{l-1}(\boldsymbol{x}_i)\right)\boldsymbol{\omega}_k$$

$$+ \frac{1}{n}\sum_{k,j}^K\sum_i^n\left(\boldsymbol{\alpha}_k^T\boldsymbol{b}_{l-1}(\boldsymbol{x}_i)\right)\boldsymbol{\omega}_k^T\boldsymbol{\omega}_j\left(\boldsymbol{\alpha}_j^T\boldsymbol{b}_{l-1}(\boldsymbol{x}_i)\right)$$

$$= -\frac{2}{n}\sum_k^K\boldsymbol{\alpha}_k^T\left(\frac{1}{n}\sum_i^n\boldsymbol{b}_{l-1}(\boldsymbol{x}_i)\boldsymbol{v}_{\text{goal}_{proj}}^{l+1}(\boldsymbol{x}_i)^T\right)\boldsymbol{\omega}_k$$

$$+ \frac{1}{n}\sum_{k,j}^K\boldsymbol{\omega}_k^T\boldsymbol{\omega}_j\boldsymbol{\alpha}_k^T\left(\frac{1}{n}\sum_i^n\boldsymbol{b}_{l-1}(\boldsymbol{x}_i)\boldsymbol{b}_{l-1}(\boldsymbol{x}_i)^T\right)\boldsymbol{\alpha}_j$$

$$= -2\sum_k^K\boldsymbol{\alpha}_k^T\boldsymbol{N}\boldsymbol{\omega}_k + \sum_{k,j}^K\boldsymbol{\omega}_k^T\boldsymbol{\omega}_j\boldsymbol{\alpha}_k^T\boldsymbol{S}\boldsymbol{\alpha}_j$$

with $\boldsymbol{N} = \frac{1}{n}\sum_i^n\boldsymbol{b}_{l-1}(\boldsymbol{x}_i)\boldsymbol{v}_{\text{goal}_{proj}}^{l+1}(\boldsymbol{x}_i)^T = \frac{1}{n}\boldsymbol{B}_{l-1}\left(\boldsymbol{V}_{\text{goal}_{proj}}^{l+1}\right)^T$,

$$\boldsymbol{S} = \frac{1}{n}\sum_i^n\boldsymbol{b}_{l-1}(\boldsymbol{x}_i)\boldsymbol{b}_{l-1}(\boldsymbol{x}_i)^T = \frac{1}{n}\boldsymbol{B}_{l-1}\boldsymbol{B}_{l-1}^T$$

Suppose $S$ is semi definite positive and note $S = S^{1/2}S^{1/2T}$, $\gamma_k = S^{1/2T}\alpha_k$ and $S^{1/2^{-1}}N = \sum_{r=1}^{R}\lambda_r v_r e_r^T$ the SVD of matrix $S^{1/2^{-1}}N$ .

$$
\begin{aligned}
-\sum_{k=1}^{K}\alpha_k^T N\omega_k &= -\sum_k \gamma_k^T S^{1/2^{-1}}N\omega_k \\
&= -\sum_k \sum_{r=1}^{R}\gamma_k^T \lambda_r v_r e_r^T \omega_k \\
&= -\operatorname{Tr}\Big(\sum_k \sum_r \lambda_r\big((\gamma_k^T v_r)e_r^T\big)\omega_k\Big) \\
&= -\operatorname{Tr}\Big(\sum_k \sum_r \lambda_r \omega_k \gamma_k^T v_r e_r^T\Big) \\
&= -\operatorname{Tr}\Big(\sum_k \omega_k \gamma_k^T \sum_r \lambda_r v_r e_r^T\Big) \\
&= -\Big\langle \sum_k \gamma_k \omega_k^T, \sum_r \lambda_r v_r e_r^T\Big\rangle_{\operatorname{Tr}} \text{ with } \langle A,B\rangle_{\operatorname{Tr}} = Trace(A^T B)
\end{aligned}
$$

$$
\begin{aligned}
\sum_{k,j}^{K}\omega_k^T \omega_j \alpha_k^T S\alpha_j &= \sum_{k,j}\big(\omega_k^T \omega_j\big)\big(\gamma_j^T \gamma_k\big) \\
&= \operatorname{Tr}\Big(\sum_{k,j}\big((\omega_k^T \omega_j)\gamma_j^T\big)\gamma_k\big)\Big) \\
&= \operatorname{Tr}\Big(\sum_{k,j}\gamma_k \omega_k^T \omega_j \gamma_j^T\Big) \\
&= ||\sum_k \omega_k \gamma_k^T||_{\operatorname{Tr}}^2 \text{ avec } ||A||_{\operatorname{Tr}} = \sqrt{Trace(A^T A)} \\
&= ||\sum_k \gamma_k \omega_k^T||_{\operatorname{Tr}}
\end{aligned}
$$

Then we have :

$$
\operatorname*{arg\,min}_{K,\hat\theta_{l\leftrightarrow l+1}}\frac{1}{n}g(\alpha,\omega) = \operatorname*{arg\,min}_{K,\alpha=S^{-1/2T}\gamma,\omega}||S^{-1/2}N - \sum_{k=1}^{K}\gamma_k \omega_k^T||_{\operatorname{Tr}}^2
$$

Then the solution is giving by the paper Eckart & Young (1936) chosing $K = rank(S^{-1/2}N)$ and $\sum_{k=1}^{K}\gamma_k \omega_k^T = \sum_{r=1}^{K}\lambda_r v_r e_r^T$. Choosing K = R is the best option. We now consider the matrix $M$.The minimization gives also the following properties at the optimum :

$$
\textbf{for } k \neq j \ \ \langle \gamma_k \omega_k^T, \gamma_j \omega_j^T\rangle_{Tr} = 0
$$

$$
||S^{-1/2}N - \sum_{k=1}^{K}\gamma_k \omega_k^T||_{\operatorname{Tr}}^2 = \sum_{r=K+1}^{R}\lambda_r^2
$$

$$
= ||S^{-1/2}N||_{Tr}^2 - ||\sum_{k=1}^{K}\gamma_k \omega_k^T||_{\operatorname{Tr}}^2
$$

Furthermore :

$$\frac{1}{n}||\boldsymbol{V}_{\text{goal}}{}^{l+1}_{proj} - \boldsymbol{V}(\hat{\theta}^{K,*}_{l\leftrightarrow l+1})||^2_{\text{Tr}} = \frac{1}{n}||\boldsymbol{V}_{\text{goal}}{}^{l+1}_{proj}||^2_{\text{Tr}} + ||(S^{1/2})^{-1}N - \sum_k \gamma_k \boldsymbol{\omega}_k^T||^2 - ||(S^{1/2})^{-1}N||^2_{\text{Tr}}$$

$$= \sum_{r=K+1}^{R} \lambda_r^2 + \frac{1}{n}||\boldsymbol{V}_{\text{goal}}{}^{l+1}_{proj}||^2_{\text{Tr}} - ||(\boldsymbol{S}^{1/2})^{-1}\boldsymbol{N}||^2_{\text{Tr}}$$

$$= -\sum_{r=1}^{K} \lambda_r^2 + \frac{1}{n}||\boldsymbol{V}_{\text{goal}}{}^{l+1}_{proj}||^2_{\text{Tr}}$$

Before looking at the impact on the Loss let's prove the corollary 3.2. We are looking for the matrix $\delta\mathbf{W}_l$ minimizing the last equality. Consider the span of matrix $S = \{\delta\mathbf{W}_l\boldsymbol{B}_l|\delta\mathbf{W}_l matrix\}$ and $\boldsymbol{P}$ the matrix projection on span $S$ :

$$\underset{\boldsymbol{D}\in S, \theta}{\arg\min} ||\boldsymbol{V}_{\text{goal}}{}^{l+1} - \boldsymbol{D} - \boldsymbol{V}^{l+1}(\theta)||^2_{\text{Tr}} = \underset{\boldsymbol{D}\in S}{\arg\min} ||\boldsymbol{P}\boldsymbol{V}_{\text{goal}}{}^{l+1} - \boldsymbol{D}||^2_{\text{Tr}}$$
$$+ \underset{\theta}{\arg\min} ||(Id - \boldsymbol{P})(\boldsymbol{V}_{\text{goal}}{}^{l+1} - \boldsymbol{V}(\theta)||^2_{\text{Tr}}$$

Then $\delta\boldsymbol{W}^*_{l+1}\boldsymbol{B}_l$ at proposition **3.1** minimize the first norm and is the orthogonal projection of $\boldsymbol{V}_{\text{goal}}{}^{l+1}$ on span $S$.

$$(Id - P)\boldsymbol{V}_{\text{goal}}{}^{l+1} = \boldsymbol{V}_{\text{goal}}{}^{l+1} - \delta\boldsymbol{W}^*_{l+1}\boldsymbol{B}_l$$

We also have that :

$$\left\langle \boldsymbol{V}^{l+1}(\delta\boldsymbol{W}^*_{l+1}), \boldsymbol{V}^{l+1}(\hat{\theta}^{K,*}_{l\leftrightarrow l+1}) \right\rangle_{\text{Tr}} = 0$$

And by definition of the orthogonal projection on linear span:

$$\left\langle \boldsymbol{V}^{l+1}(\theta^{K,*}_{l\leftrightarrow l+1}), \boldsymbol{V}_{\text{goal}}{}^{l+1} \right\rangle_{\text{Tr}} = \left\langle \boldsymbol{V}^{l+1}(\theta^{K,*}_{l\leftrightarrow l+1}), \boldsymbol{V}_{\text{goal}}{}^{l+1}_{proj}(\delta\boldsymbol{W}^*_{l+1}) \right\rangle_{\text{Tr}}$$

We note $\boldsymbol{V}_{\text{goal}}{}^{l+1}_{proj}(\delta\boldsymbol{W}^*_{l+1}) := \boldsymbol{V}_{\text{goal}}{}^{l+1}_{proj}$ The impact on the global loss is :

$$\frac{1}{n}\sum_{i=1}^{n}\mathcal{L}(f_{\theta(\gamma_0^*\delta\boldsymbol{W}^*_{l+1},l+1)\oplus(\hat{\boldsymbol{\alpha}}^*_k,\hat{w}^*_k)^R_{k=1}(\gamma)}(\boldsymbol{x}_i), \boldsymbol{y}_i)$$

$$= \frac{1}{n}\sum_{i=1}^{n}\left(L(f_\theta(\boldsymbol{x}_i), \boldsymbol{y}_i) + \sigma'_l(0)\left\langle \nabla_{\boldsymbol{a}_{l+1}(\boldsymbol{x}_i)}L(f_\theta(\boldsymbol{x}_i), \boldsymbol{y}_i), \boldsymbol{v}^{l+1}(\boldsymbol{x}_i, (\theta^{K,*}_{l\leftrightarrow l+1}(\gamma), \gamma_0\delta\boldsymbol{W}^*_{l+1}) \right\rangle\right)$$
$$+ o(max(\gamma, \gamma_0))$$

$$' = \frac{1}{n}\sum_{i=1}^{n}L(f_\theta(\boldsymbol{x}_i), \boldsymbol{y}_i) - \frac{\gamma}{\eta}\frac{1}{n}\sigma'_l(0)\left\langle \boldsymbol{V}^{l+1}(\theta^{K,*}_{l\leftrightarrow l+1}), \boldsymbol{V}_{\text{goal}}{}^{l+1} \right\rangle_{\text{Tr}} - \frac{1}{\eta}\frac{1}{n}\sigma'_l(0)\left\langle \boldsymbol{V}^{l+1}(\delta\gamma_0\boldsymbol{W}^*_{l+1}), \boldsymbol{V}_{\text{goal}}{}^{l+1} \right\rangle_{\text{Tr}}$$
$$+ o(max(\gamma, \gamma_0))$$

$$= \frac{1}{n}\sum_{i=1}^{n}L(f_\theta(\boldsymbol{x}_i), \boldsymbol{y}_i) - \frac{\gamma}{\eta}\frac{1}{n}\sigma'_l(0)\left\langle \boldsymbol{V}^{l+1}(\theta^{K,*}_{l\leftrightarrow l+1}), \boldsymbol{V}_{\text{goal}}{}^{l+1}_{proj} \right\rangle_{\text{Tr}} - \frac{\gamma_0}{\eta}\frac{1}{n}\sigma'_l(0)\left\langle \boldsymbol{V}^{l+1}(\delta\boldsymbol{W}^*_{l+1}), \boldsymbol{V}_{\text{goal}}{}^{l+1} \right\rangle_{\text{Tr}}$$
$$+ o(max(\gamma, \gamma_0))$$

We also have the following property :

$$\underset{\hat{\theta}^K_{l\leftrightarrow l+1}}{\arg\min} \left\{ \frac{1}{n} ||\boldsymbol{V}_{\text{goal}\,proj}^{l+1} - \boldsymbol{V}^{l+1}(\hat{\theta}^K_{l\leftrightarrow l+1})||^2_{\text{Tr}} \right\}$$

$$= \underset{H\geq 0}{\arg\min} \quad \underset{\hat{\theta}^K_{l\leftrightarrow l+1}, ||\boldsymbol{V}^{l+1}(\hat{\theta}^K_{l\leftrightarrow l+1})||_{\text{Tr}}=H}{\arg\min} \left\{ \frac{1}{n}||\boldsymbol{V}_{\text{goal}\,proj}^{l+1} - \boldsymbol{V}^{l+1}(\hat{\theta}^K_{l\leftrightarrow l+1}, \delta\boldsymbol{W}^*_{l+1})||^2_{\text{Tr}} \right\}$$

$$= \underset{H\geq 0}{\arg\min} \quad \underset{\hat{\theta}^K_{l\leftrightarrow l+1}, ||\boldsymbol{V}^{l+1}(\hat{\theta}^K_{l\leftrightarrow l+1})||_{\text{Tr}}=H}{\arg\min} \left\{ -\frac{2}{n}\left\langle \boldsymbol{V}_{\text{goal}\,proj}^{l+1}(, \boldsymbol{V}^{l+1}(\hat{\theta}^K_{l\leftrightarrow l+1})\right\rangle_{\text{Tr}} + \frac{1}{n}||\boldsymbol{V}_K(\hat{\theta}^K_{l\leftrightarrow l+1})||^2_{\text{Tr}} \right\}$$

$$= \underset{H\geq 0}{\arg\min} \quad \underset{\hat{\theta}^K_{l\leftrightarrow l+1}, ||\boldsymbol{V}^{l+1}(\hat{\theta}^K_{l\leftrightarrow l+1})||_{\text{Tr}}=H}{\arg\min} \left\{ -\frac{2}{n}\left\langle \boldsymbol{V}_{\text{goal}\,proj}^{l+1}, \boldsymbol{V}^{l+1}(\hat{\theta}^K_{l\leftrightarrow l+1})\right\rangle_{\text{Tr}} + \frac{1}{n}H^2 \right\}$$

$$= \underset{H\geq 0}{\arg\min} \quad \underset{\hat{\theta}^K_{l\leftrightarrow l+1}, ||\boldsymbol{V}^{l+1}(\hat{\theta}^K_{l\leftrightarrow l+1})^*||_{\text{Tr}}=1}{\arg\min} \left\{ -H\left\langle \boldsymbol{V}_{\text{goal}\,proj}^{l+1}, \boldsymbol{V}^{l+1}(\hat{\theta}^K_{l\leftrightarrow l+1})\right\rangle_{\text{Tr}} + \frac{1}{2}H^2 \right\}$$

With $\boldsymbol{V}^{l+1}(\hat{\theta}^K_{l\leftrightarrow l+1})^*$ the solution of the second arg min (ie for $H = 1$).
Then the norm minimizing the first argmin is given by :

$$H^* = \left\langle \boldsymbol{V}_{\text{goal}\,proj}^{l+1}, \boldsymbol{V}^{l+1}(\hat{\theta}^K_{l\leftrightarrow l+1})^*\right\rangle_{\text{Tr}}$$

$$= \left\langle \boldsymbol{V}_{\text{goal}}^{l+1}, \boldsymbol{V}^{l+1}(\hat{\theta}^K_{l\leftrightarrow l+1})^*\right\rangle_{\text{Tr}} - 0$$

Furthermore

$$\underset{\hat{\theta}^K_{l\leftrightarrow l+1}}{\min} \left\{ \frac{1}{n}||\boldsymbol{V}_{\text{goal}\,proj}^{l+1} - \boldsymbol{V}^{l+1}(\hat{\theta}^K_{l\leftrightarrow l+1})||^2_{\text{Tr}} \right\} = -\sum_{r=1}^{K}\lambda_r^2 + \frac{1}{n}||\boldsymbol{V}_{\text{goal}\,proj}^{l+1}||^2_{\text{Tr}}$$

$$\underset{\hat{\theta}^K_{l\leftrightarrow l+1}}{\min} \left\{ \frac{1}{n}||\boldsymbol{V}_{\text{goal}\,proj}^{l+1} - \boldsymbol{V}^{l+1}(\hat{\theta}^K_{l\leftrightarrow l+1})||^2_{\text{Tr}} \right\} = -\frac{1}{n}H^{*2} + \frac{1}{n}||\boldsymbol{V}_{\text{goal}\,proj}^{l+1}||_{\text{Tr}}$$

$$\implies H^* = \left\langle \boldsymbol{V}_{\text{goal}\,proj}^{l+1}, \boldsymbol{V}^{l+1}(\hat{\theta}^K_{l\leftrightarrow l+1})^*\right\rangle_{\text{Tr}} = \sqrt{\sum_{r=1}^{K}\lambda_r^2} \times \sqrt{n}$$

$$\boldsymbol{V}^{l+1}(\hat{\theta}^{K,*}_{l\leftrightarrow l+1}) = H^*\boldsymbol{V}^{l+1}(\hat{\theta}^K_{l\leftrightarrow l+1})^*$$

$$\left\langle \boldsymbol{V}^{l+1}(\theta^{K,*}_{l\leftrightarrow l+1}), \boldsymbol{V}_{\text{goal}\,proj}^{l+1}\right\rangle_{\text{Tr}} = H^* \times \left\langle \boldsymbol{V}_{\text{goal}\,proj}^{l+1}, \boldsymbol{V}^{l+1}(\hat{\theta}^K_{l\leftrightarrow l+1})^*\right\rangle_{\text{Tr}} = H^{*2}$$

where the last equality is given by the optimisation of $||\boldsymbol{S}^{-1/2}\boldsymbol{N} - \sum_{k=1}^{K}\boldsymbol{u}_k\boldsymbol{\omega}_k^T||^2_{Tr}$. So minimizing the scalar product $-\left\langle \boldsymbol{V}_{\text{goal}\,proj}^{l+1}(\delta\boldsymbol{W}^*_{l+1}), \boldsymbol{V}^{l+1}(\hat{\theta}^K_{l\leftrightarrow l+1})^*\right\rangle_{\text{Tr}}$ setting the norm of $\boldsymbol{V}^{l+1}(\hat{\theta}^K_{l\leftrightarrow l+1})$ is equivalent to minimizing the norm $||\boldsymbol{V}_{\text{goal}\,proj}^{l+1}(\delta\boldsymbol{W}^*_{l+1}) - \boldsymbol{V}^{l+1}(\hat{\theta}^K_{l\leftrightarrow l+1})||^2_{\text{Tr}}$.

$$\frac{1}{n}\sum_{i=1}^{n}\mathcal{L}(f_{\theta(\gamma_0^*\boldsymbol{W}^*_{l+1})\oplus(\hat{\boldsymbol{\alpha}}_k^{l+1,*}, \hat{\boldsymbol{\omega}}_k^*)_{k=1}^R}(\gamma)(\boldsymbol{x}_i), \boldsymbol{y}_i)$$

$$= \frac{1}{n}\sum_{i=1}^{n}L(f_\theta(\boldsymbol{x}_i), \boldsymbol{y}_i) - \frac{\gamma}{\eta}\sigma_l'(0)\sum_{r=1}^{K}\lambda_k^2 - \frac{\gamma_0}{\eta}\frac{1}{n}\sigma_l'(0)\langle\boldsymbol{V}^{l+1}(\boldsymbol{W}^*_{l+1}), \boldsymbol{V}_{\text{goal}}^{l+1}\rangle_{\text{Tr}} + o(max(\gamma, \gamma_0))$$

### B.3 CHOLESKY DECOMPOSITION FOR S POSITIVE SEMI-DEFINITE

When matrix $\boldsymbol{S}$ is not positive definite, the following trick can be apply. Consider $\boldsymbol{S} = \boldsymbol{U}\Sigma\boldsymbol{V}^T$ the svd of $\boldsymbol{S}$. As $\boldsymbol{S}$ is symmetric $\boldsymbol{U} = \boldsymbol{V}^T$. Perform the QR-decomposition of matrix $\sqrt{\Sigma}\boldsymbol{U}^T = \boldsymbol{Q}\boldsymbol{R}$, $\boldsymbol{Q}$ an orthogonal matrix and $\boldsymbol{R}$ an upper triangular matrix. Defining $\boldsymbol{P}^+$ the pseudo inverse of $\boldsymbol{P}$, one can remark that $\boldsymbol{R}^T\boldsymbol{R} = \boldsymbol{U}\sqrt{\Sigma}(\boldsymbol{Q}^{-1})^T\boldsymbol{Q}^{-1}\boldsymbol{U}\sqrt{\Sigma}\boldsymbol{U}^T$. As $\boldsymbol{Q}$ is orthogonal $\boldsymbol{R}^T\boldsymbol{R} = \boldsymbol{U}\Sigma\boldsymbol{U}^T = \boldsymbol{U}\Sigma\boldsymbol{V}^T = \boldsymbol{S}$.

### B.4 PROPOSITION 3.3

Suppose $S$ is semi definite, we note $S = S^{1/2}S^{1/2T}$. Solving (7) is equivalent to taking $\omega_k = N^T\alpha_k$ and find the K first eigenvectors $\omega_k$ associated to the K largest eigenvalues $\lambda$ of the generalized eigenvalue problem :

$$NN^T\alpha_k = \lambda S\alpha_k$$

*Proof* The LOBPCG problem is equivalent to maximise the generalized Rayleigh quotient which is :

$$\alpha^* = max_x \frac{\alpha^T NN^T \alpha}{\alpha^T S\alpha}$$

$$p^* = max_{p=S^{1/2T}\alpha} \frac{p^T S^{1/2^{-1}} NN^T S^{1/2^{-1T}} p}{p^T p}$$

$$p^* = max_{||p||=1} ||N^T S^{1/2^{-1T}} p||$$

$$\alpha^* = S^{1/2^{-1T}} p^*$$

And considering the SVD of $S^{1/2^{-1}} N = \sum_{r=1}^{R} \lambda_r u_r v_r^T$, then $S^{1/2^{-1}} NN^T S^{1/2^{-1T}} = \sum_{r=1}^{R} \lambda_r^2 u_r u_r^T$ because $j \neq i \implies u_i^T u_j = 0$ and $v_i^T v_j = 0$. So maximise the first formula is equivalent to $p_k^* = u_k$, then $\alpha_k = S^{1/2^{-1T}} u_k$. And $N^T\alpha_k = N^T S^{1/2^{-1T}} u_k = \lambda_k v_k$

We prove second corollary 3.2 by induction. For $m = m' = 1$ :

$$a_{l+1}(x)^{t+1} = a_{l+1}(x)^t + V(\hat{\theta}_{l\leftrightarrow l+1}^{1,*}, x)\gamma + o(\gamma)$$

$$v_{\text{goal}}^{l+1,t+1}(x) = v_{\text{goal}}^{l+1,t}(x) + \nabla_{a_{l+1}(x)}\mathcal{L}(f_\theta(x), y)^T v(\hat{\theta}_{l\leftrightarrow l+1}^{1,*}, x)\gamma + o(\gamma)$$

Adding the second neuron we obtain the minimization problem:

$$\arg\min_{\alpha_2,\omega_2} ||V_{\text{goal}}^{l+1,t} - V^{l+1}(\alpha_2, \omega_2)||_{\text{Tr}} + o(1)$$

### B.5 SECTION *Theory behind Greedy Growth* WITH PROOFS

One might wonder whether a greedy approach on layer growth might get stuck in a non-optimal state. We derive the following series of propositions in this regard. Since in this work we add neurons layer per layer independently, we study here the case of a single hidden layer network, to spot potential layer growth issues. For the sake of simplicity, we consider the task of least square regression towards an explicit continuous target $f^*$, defined on a compact set. That is, we aim at minimizing the loss:

$$\inf_f \sum_{x \in \mathcal{D}} \|f(x) - f^*(x)\|^2$$

where $f(x)$ is the output of the neural network and $\mathcal{D}$ is the training set.

**Proposition B.1** (Greedy completion of an existing network)**.** *If $f$ is not $f^*$ yet, there exists a set of neurons to add to the hidden layer such that the new function $f'$ will have a lower loss than $f$.*

One can even choose the added neurons such that the loss is arbitrarily well minimized.

*Proof.* The classic universal approximation theorem about neural networks with one hidden layer Pinkus (1999) states that for any continuous function $g$ defined on a compact set $\omega$, for any desired precision $\gamma$, and for any activation function $\sigma$ provided it is not a polynomial, then there exists a neural network $\hat{g}$ with one hidden layer (possibly quite large when $\gamma$ is small) and with this activation function $\sigma$, such that

$$\forall x, \|g(x) - g^*(x)\| \leqslant \gamma$$

We apply this theorem to the case where $g^* = f^* - f$, which is continuous as $f^*$ is continuous, and $f$ is a shallow neural network and as such is a composition of linear functions and of the function $\sigma$, that we will suppose to be continuous for the sake of simplicity. We will suppose that $f$ is real-valued for the sake of simplicity as well, but the result is trivially extendable to vector-valued functions (just

concatenate the networks obtained for each output independently). We choose $\gamma = \frac{1}{10}\|f^* - f\|_{L2}$, where $\langle a | b \rangle_{L2} = \frac{1}{|\boldsymbol{\omega}|} \int_{x \in \boldsymbol{\omega}} a(x) b(x) \, dx$. This way we obtain a one-hidden-layer neural network $g$ with activation function $\sigma$ such that:

$$\forall x \in \boldsymbol{\omega}, \quad -\gamma \leqslant g(x) - g^*(x) \leqslant \gamma$$

$$\forall x \in \boldsymbol{\omega}, \quad g(x) = f^*(x) - f(x) + a(x)$$

with $\forall x \in \boldsymbol{\omega}, \ |a(x)| \leqslant \gamma$.

Then:

$$\forall x \in \boldsymbol{\omega}, \quad f^*(x) - (f(x) + g(x)) = -a(x)$$

$$\forall x \in \boldsymbol{\omega}, \quad (f^*(x) - h(x))^2 = a^2(x) \tag{12}$$

with $h$ being the function corresponding to a neural network consisting in concatenating the hidden layer neurons of $f$ and $g$, and consequently summing their outputs.

$$\|f^* - h\|_{L2}^2 = \|a\|_{L2}^2$$

$$\|f^* - h\|_{L2}^2 \leqslant \gamma^2 = \frac{1}{100}\|f^* - f\|_{L2}^2$$

and consequently the loss is reduced indeed (by a factor of 100 in this construction).

The same holds in expectation or sum over a training set, by choosing $\gamma = \frac{1}{10}\sqrt{\frac{1}{|\mathcal{D}|}\sum_{x \in \mathcal{D}}\|f(x) - f^*(x)\|^2}$, as Equation (12) then yields:

$$\sum_{x \in \mathcal{D}}(f^*(x) - h(x))^2 = \sum_{x \in \mathcal{D}} a^2(x) \leqslant \frac{1}{100}\sum_{x \in \mathcal{D}}(f^*(x) - f(x))^2$$

which proves the proposition as stated.

For more general losses, one can consider order-1 (linear) developpment of the loss and ask for a network $g$ that is close to (the opposite of) the gradient of the loss.

$\square$

*Proof of the additional remark.* The proof in Pinkus (1999) relies on the existence of real values $c_n$ such that the $n$-th order derivatives $\sigma^{(n)}(c_n)$ are not 0. Then, by considering appropriate values arbitrarily close to $c_n$, one can approximate the $n$-th derivative of $\sigma$ at $c_n$ and consequently the polynomial $c^n$ of order $n$. This standard proof then concludes by density of polynomials in continuous functions.

Provided the activation function $\sigma$ is not a polynomial, these values $c_n$ can actually be chosen arbitrarily, in particular arbitrarily close to 0. This corresponds to choosing neuron input weights arbitrarily close to 0. $\square$

**Proposition B.2** (Greedy completion by one single neuron). *If $f$ is not $f^*$ yet, there exists a neuron to add to the hidden layer such that the new function $f'$ will have a lower loss than $f$.*

*Proof.* From the previous proposition, there exists a finite set of neurons to add such that the loss will be decreased. In this particular setting of $L2$ regression, or for more general losses if considering small function moves, this means that the function represented by this set of neurons has a strictly negative component over the gradient $g$ of the loss ($g = 2(f^* - f)$ in the case of the $L2$ regression). That is, denoting by $a_i\sigma(\boldsymbol{W}_i \cdot \boldsymbol{x})$ these $N$ neurons:

$$\Big\langle \sum_{i=1}^{N} a_i\sigma(\boldsymbol{w}_i \cdot \boldsymbol{x}) \,\Big|\, g \Big\rangle_{L2} = K < 0$$

i.e.

$$\sum_{i=1}^{N} \langle a_i\sigma(\boldsymbol{w}_i \cdot \boldsymbol{x}) |\, g \rangle_{L2} = K < 0$$

Now, by contradiction, if there existed no neuron $i$ among these ones such that

$$\langle a_i \sigma(\boldsymbol{w}_i \cdot \boldsymbol{x}) \mid g \rangle_{L2} \leqslant \frac{1}{N} K$$

then we would have:

$$\forall i \in [1, N], \;\; \langle a_i \sigma(\boldsymbol{w}_i \cdot \boldsymbol{x}) \mid g \rangle_{L2} > \frac{1}{N} K$$

$$\sum_{i=1}^{N} \langle a_i \sigma(\boldsymbol{w}_i \cdot \boldsymbol{x}) \mid g \rangle_{L2} > K$$

hence a contradiction. Then necessarily at least one of the $N$ neurons satisfies

$$\langle a_i \sigma(\boldsymbol{w}_i \cdot \boldsymbol{x}) \mid g \rangle_{L2} \leqslant \frac{1}{N} K < 0$$

and thus decreases the loss when added to the hidden layer of the neural network representing $f$. Moreover this decrease is at least $\frac{1}{N}$ of the loss decrease resulting from the addition of all neurons.

$\square$

As a consequence, our greedy approach will not get stuck in a situation where one would need to add many neurons simultaneously to decrease the loss: it is always feasible by a single neuron. On can express a lower bound on how much the loss has improved (for the best such neuron), but it not a very good one without further assumptions on $f$.

**Proposition B.3** (Greedy completion by one infinitesimal neuron). *The neuron in the previous proposition can be chosen to have arbitrarily small input weights.*

*Proof.* This is straightforward, as, following a previous remark, the neurons found to collectively decrease the loss can be supposed to all have arbitrarily small input weights. $\square$

This detail is important in that our approach is based on the tangent space of the function $f$ and consequently manipulates infinitesimal quantities. Though we perform line search in a second step and consequently add non-infinitesimal neurons, our first optimization problem relies on the linearization of the activation function by requiring the added neuron to have infinitely small input weights, without which it would be much harder to solve. This proposition confirms that such neuron does exist indeed.

## C   ADDITIONAL EXPERIMENTAL RESULTS

All the experiements are performed 20 times on MNIST Dataset and the models are trained with $Adam(lr = 0.0001, \mu = 0, batchsize = 100)$ with 7 CPU. For the classic model, neurons are initialized with Kaiming Normal. For our approach, we always start with a model of size $[1, 1]$ initialized with Kaiming Normal and we expand its architecture using 3.2 every 0.05 seconds. The batch-size, $n_{mb}^l$, for estimating $\theta(\gamma)_{l \leftrightarrow l+1}(\gamma)$ and $\delta W_{l+1}^*$ is $n_{mb}^l = 100$ at first step and for every layer. After applying our method on every layers, $n_{mb}$ is updated as $n_{mb}^l \leftarrow n_{mb}^l \times \sqrt{max(neurons_{l-1}, neurons_{l+1})}$, where $neurons_l$ indicates the amount of neurons at layer $l$. In the following section, we modify the value for the training time between architecture growth 8 and the architecture growth 10. The $y - axis$ is accuracy on test set.

### C.1   CHANGING TRAINING TIME BETWEEN EACH ARCHITECTURE EXPANSION STEP

In this part we modify the training time between each architecture growth. We apply 8 times our method on each hidden layer, the final architecture is $[222, 71]$

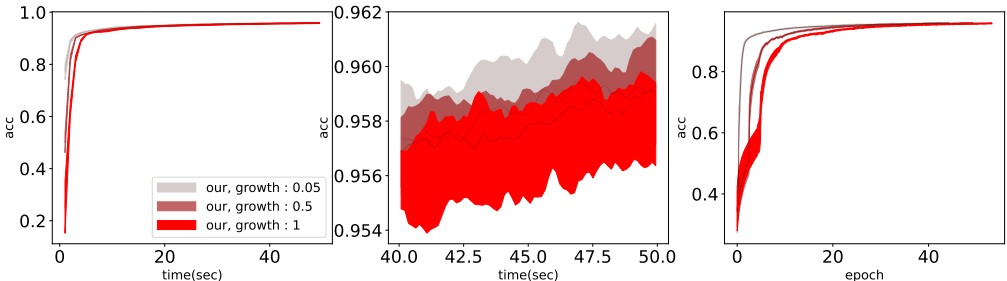

Figure 8: The different shades of red correspond to different training time between each architectures growth in seconds.

### C.2   SMALL ARCHITECTURE

Same plot as 6 but with others architecture.

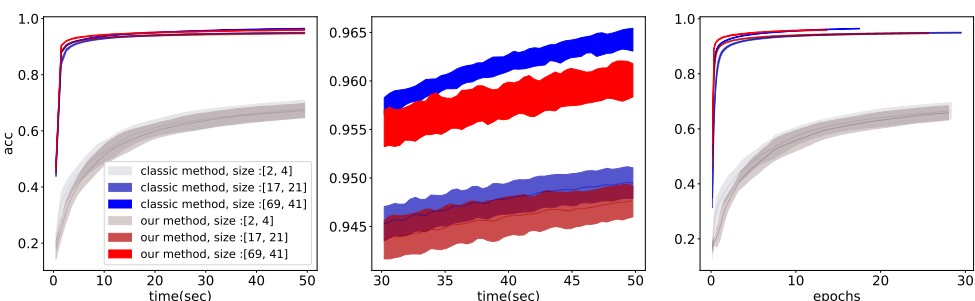

Figure 9: Right interpolation on interval of size 0.01. Middle : accuracy as a function of mini-batches. Right : accuracy plotted as a function of the mean of time in each category : classic training and our approach.

### C.3   EIGENVALUES

In this section we apply our method 15 times on each layer. After applying 8 times our method, the accuracy of the system does not increase significantly. We have that $\mathbb{E}_{x,y \sim \mathcal{D}}[z^T v_{\text{goal}_l}(x)]$ for every

vector $\boldsymbol{z}$, as explained in part 5. Looking at the eigenvalues for the first and second hidden layers and

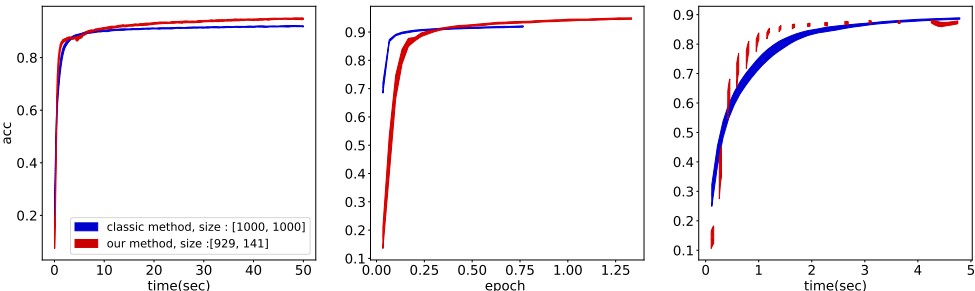

Figure 10: Comparison between the classic training of architecture $[1000, 1000]$ and our approach of architecture $[929, 141]$. All models are trained for 20 seconds we plot here mean and standard deviation for accuracy on test set. For our method we trained the model for 0.05 seconds between each architecture growth. Right interpolation on interval of size 0.01. Middle : accuracy as a function of epochs. Right : accuracy plotted as a function of the mean of time in each category : classic training and our approach.

estimate decrease for the loss, ie $\sum_k \lambda_k^2$ :

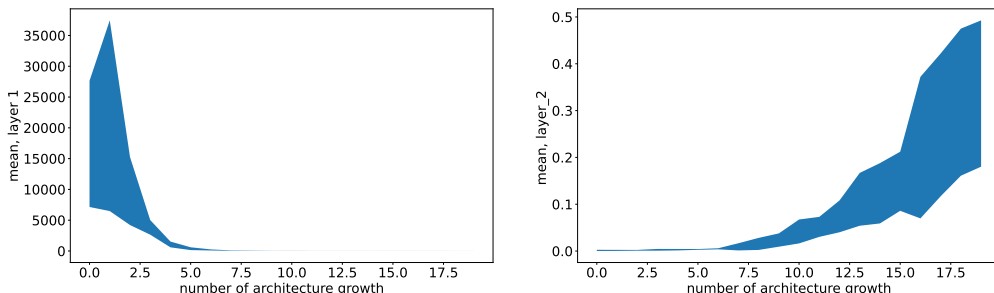

Figure 11: mean and standard deviation for $\frac{1}{k}\sum_k \lambda_k^2$ for first and second hidden layers as a function of the number of architecture growths

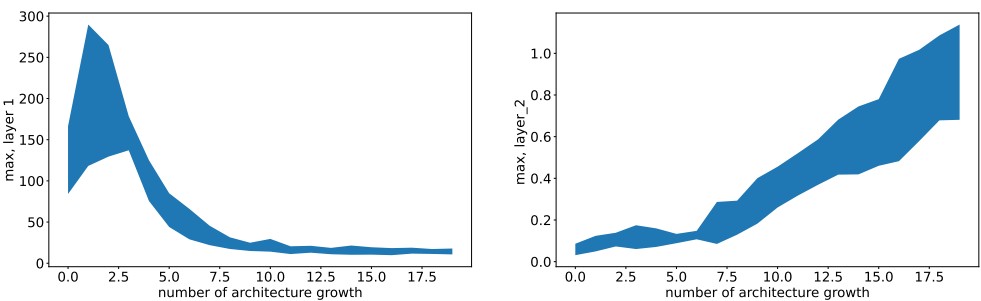

Figure 12: mean and standard deviation for $max(\lambda_k)$ for first and second hidden layers as a function of the number of architecture growths

## C.4 RANDOM VS OPTIMIZATION

When performing the quadratic optimization (5), we obtain the optimal direction for $(\hat{\alpha}_k^*, \hat{\omega}_k^*)_{k=1}^R$. It is also possible to generate randomly the new neurons and compute the amplitude factors. This second option have the benefit of being less time consuming, but it would project the desired direction on those random vectors and would affect the accuracy score compared to optimal solution defined in 3.1.

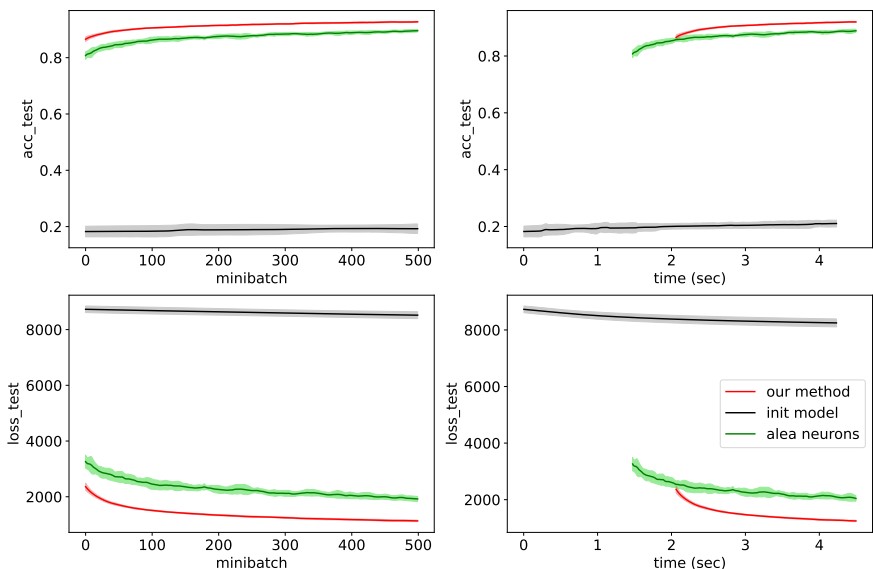

Figure 13: Experiment performed 20 times on the MNIST dataset: a starting model in black $[1, 1]$ is initialized according to normal Kaiming, then is duplicated to give the red and the green model. The structure of the red model is modified by our method to reach the structure $[110, 51]$ while the green model is extended with random neurons. Then all models are trained for 5 seconds. The white space for the red model corresponds to the quadratic optimisation and the computation of the amplitude factor while for the green model it corresponds only to the computation of the amplitude factor.

