# OpenReview forum: "Spotting Expressivity Bottlenecks and Fixing Them Optimally "
_ICLR.cc/2023/Conference — Submitted to ICLR 2023_

### Official Review · Reviewer_ae2U · 2022-10-23

**Confidence:** 4
**Correctness:** 3
**Technical Novelty And Significance:** 2
**Empirical Novelty And Significance:** 1
**Recommendation:** 3

**Clarity, Quality, Novelty And Reproducibility:**

The paper is well written and the derivation was easy to follow. The originality of the work may be limited, as the idea is quite similar to or not as novel as prior works, e.g., Wu et al., 2019.

**Strength And Weaknesses:**

Strength:
- First of all, I think the problem of growing neural networks on the fly is very interesting and important, especially putting it under continual/lifelong learning scenarios.
- In terms of the methodology of this paper, I enjoyed reading the motivation and the derivation of the method, which I feel is pretty natural. In particular, I like the toy example for illustrating the issue of lacking expressivity.

Weakness:
- The most critical weakness of this paper is that the authors seem to miss many important related kinds of literature, and failed to make a comparison with them. To my knowledge, the problem of growing neural networks on the fly is not new. There have already been many works.
 - - For example, the work by Wu et al., 2019, is very similar to your idea. The high-level idea is that adding more neurons will turn local optima in the small neural networks to be a saddle point. They derive a simple criterion for deciding the best subset of neurons to split and a splitting gradient for optimally updating the off-springs, which is a second-order functional steepest descent for escaping saddle points in an $\infty$-Wasserstein metric space. There are also follow-up works by Wu et al., 2020 and Wang et al., 2019.
- - The work by Chen et al., 2015, is a bit more empirical but is also very relevant. In their work, they add new neurons with carefully designed weights that don't change the computed function. The only difference is that they need to manually decide when to add neurons.
Therefore, I believe a detailed discussion and comprehensive comparisons of these methods are necessary.

There are also a couple of additional concerns:
- The derivation in the paper is only about one layer. How can it generalize to multiple layers?
- How do you decide the number of neurons be added? Or do you need to enumerate all the possibilities?
- The experiments are quite lacking. I would suggest the author follow the experiment setup in [1-4].


References:

[1] Wu, Lemeng, Dilin Wang, and Qiang Liu. "Splitting steepest descent for growing neural architectures." Advances in neural information processing systems 32 (2019).

[2] Wu, Lemeng, Mao Ye, Qi Lei, Jason D. Lee, and Qiang Liu. "Steepest descent neural architecture optimization: Escaping local optimum with signed neural splitting." arXiv preprint arXiv:2003.10392 (2020).

[3] Wang, D., Li, M., Wu, L., Chandra, V., & Liu, Q. (2019). Energy-aware neural architecture optimization with fast splitting steepest descent. arXiv preprint arXiv:1910.03103.

[4] Chen, Tianqi, Ian Goodfellow, and Jonathon Shlens. "Net2net: Accelerating learning via knowledge transfer." ICLR 2016.

**Summary Of The Paper:**

This paper proposes a method for growing neural networks during training. The authors formulate the problem as an optimization problem. Specifically, when the network lacks expressivity, there will be no gradient updates in the weight space that will lead to a change in the outputs that align with the desired functional gradient, i.e., $v_{goal}$. To mitigate this issue, the authors propose to expand the network by adding extra neurons and assigning the weights by solving an optimization problem detailed in section 3. Empirically, the authors conduct preliminary experiments on MNIST datasets.


**Summary Of The Review:**

Overall, I think the problem studied in this paper is interesting and important. The motivation is clear and well-justified. However, my major concern is that the authors missed many related literatures and hence failed to make a comparisons with them. The position of the paper is unclear. Detailed comments can be found in Strength And Weaknesses section.

---

> ### Author Response · Authors · 2022-11-18
> **About multilayer and number of added neurons**
>
> We have replied to the literature and experiment concerns in our general answer to all reviewers. Regarding specific questions:
>
>
> _The derivation in the paper is only about one layer. How can it generalize to multiple layers?_
>
> The framework demonstrated in the paper to find and add the best neurons to a given layer is actually sequentially applied to each layer. If not using the line search (to find the optimal amplitude of the added neuron), the added neurons are infinitesimal (not changing the function but only its derivative) and consequently the addition to every layer can be done independently in parallel. Even if using the line-search, one could actually perform additions in parallel over sets of non-consecutive layers.
>
>
> _How do you decide the number of neurons be added? Or do you need to enumerate all the possibilities?_
>
> The eigenvalues given by the SVD are very informative in this respect. As stated in Proposition 3.2, the expressivity gain, which turns out to be equal to the loss gain, is just the addition of the eigenvalues related to the added neurons. So one can easily design an addition strategy based on the values of these eigenvalues (for instance, the ones above some threshold, and within a maximum number). One does actually not need to perform the SVD to know the total possible gain (if adding all possible neurons) as this is the trace of the matrix, which is straightforwadly computable. So, one could drop the SVD computation if no significant gain is expected.

---

### Official Review · Reviewer_vKnn · 2022-10-30

**Confidence:** 4
**Correctness:** 4
**Technical Novelty And Significance:** 3
**Empirical Novelty And Significance:** 2
**Recommendation:** 3

**Clarity, Quality, Novelty And Reproducibility:**

- The paper was relatively clear to read. There were several minor details that were difficult to parse as well as some sections that could benefit from more elaboration.
- The quality of the paper is high in the technical math content. However, left a lot to be desired in the experimental content.
- To my knowledge the approach considered in this paper is novel.
- I did not attempt to reproduce the results of this paper.

# Clarity issues
- It is not clear to me how the first equation in the gradient descent reminder turns into the latter part of the second one.
- In section 2.3 Ideal updates, backbone propagation should be backpropagation and preactivities should be preactivations. Also $a^l$ should be $a_l$.

**Strength And Weaknesses:**

# Strengths
- The paper is very thorough and clearly describes all the math involved in their approach.
- To my knowledge, the paper presents a novel approach for extending neural networks using functional gradient information.
- The suggested method for extending neural networks seems very powerful if works as claimed and could have a large impact on the way we train networks.

# Weaknesses
- One hypothesis why Deep Learning works so well is that the fixed architecture provides an implicit regularization that ensures the network doesn't overfit while SGD and variants optimize the training loss. The approach presented in this paper removes this form of implicit regularization.
While this approach may work, the generalization of this method needs to be validated through experiments.
The only experiments conducted in this paper are small-scale experiments on MNIST using impoverished networks.
MNIST is too small of a dataset to suffer the actual generalization consequences that may come from using this method. I would like to see experiments on larger datasets like CIFAR-10 with larger overparameterized baselines as suggested by most of the contemporary deep learning literature.
- Most contemporary deep learning models are overparameterized, meaning they are able to achieve 100 train accuracy without needing to add any more parameters. While there may be an expressivity gap at each training step for these models, there does not seem to be any expressivity gap for the actual task since the models can easily overfit on the training set. This suggests that the key assumption made in this paper regarding the expressivity gap is flawed.
- I would like to see other experiments comparing this method to alternative methods like pruning that also attempt to identify "lottery ticket" architectures. If the method proposed here achieves a similar generalization metric to those methods, I would be more convinced of the claims made by the paper.
- This approach is analogous to taking a low-degree polynomial and increasing its order to fit the data. While the approach considered is novel considering the way the network is extended using the gradient information, the final success of this approach will lie entirely on its ultimate generalization performance for which the current experiments are not convincing enough.
- It's unclear why the experiments use time on the x-axis.
- Primarily my objections boil down to the limitations of using MNIST to evaluate an architectural search problem and the use of small networks that do not express the interpolation regime behavior to which we attribute the success of deep learning models.


**Summary Of The Paper:**

The paper suggests a novel approach to extend neural networks during training using the functional gradient. The approach starts with an impoverished network and iteratively adds neurons to the network making sure to follow the true gradient direction. The authors mathematically derive an optimal technique to grow the architecture in the manner proposed. They also present experiments on MNIST showing that their approach performs better than classically trained networks with similar architectural constraints and networks extended by randomly adding neurons.

**Summary Of The Review:**

While the contributions of this paper are novel to my knowledge and the math describing the motivations and details of the approach is extensive. The hypothesis of this paper remains experimentally unverified in my opinion (See weaknesses section). As a result, without more experiments, I don't think this paper is ready for publication in its current form.

---

> ### Author Response · Authors · 2022-11-18
> **Specific comments**
>
> We have replied to the literature and experiment concerns in our general answer to all reviewers. Regarding the specific questions:
>
> _About overparameterized networks:_
>
> We agree that overparameterized networks, which are the standard in common practice, do not suffer from expressivity bottlenecks. Let us formulate this differently: people use overparameterized networks in daily practice, because these train well, while small networks do not (to the price of having to reduce the model afterwards). Small networks do not train well, because they suffer from expressivity and optimization issues, ie. many local minima (that disappear when considering plenty of neurons). So if one was about to solve these optimization and expressivity issues of small networks, people might consider training small networks again instead of large ones. Hence the point of being able to follow the functional gradient.
>
> _About implicit regularization and generalization_
>
> This is a very good point. Classical machine learning always adds regularizers to the criterion to be optimized, while this wise practice has been lost with the advent of deep learning. If one forgets to consider regularizers, and follows the functional gradient descent till the end, overfitting will show up in quantities. We therefore advocate for considering regularizers again, as the expert architecture design (obtained experimentally with a lot of trials) will not be there anymore to play this role. This is expected as one gets closer to the real functional gradient descent. We were aware of this potential issue, but after further thoughts, the information about potential overfitting might actually lie in the SVD we compute. Indeed, one can check the improvements brought by an eigenvector / new neuron, more precisely we can check the number of samples for which its addition will improve the loss. If too few, risk of overfitting; if many, no risk. A good adding strategy could be based on this. Further work is needed to develop such ideas. However in practice we noticed that so far during our basic experiments, our models do not overfit; this is simply due to the very small number of parameters in the final architecture, which prevents overfitting. So further work on larger scales is needed to check whether and how overfitting happens, and whether simple strategies such as the one above are needed and work or not.
>
> _About lottery tickets_
>
> The issue with "lottery tickets" methods is that they rely on combinatorics: the larger the network, the higher the chances to have from the beginning random neurons that do the job. These are kind of brute force methods, wasting computational resources. We precisely try to avoid them, or to use them wisely only when and where really needed, that is, to use the power of such non-linear random projections only when no other information is exploitable, as suggested in Section 5. While we agree that the theory (and experiments) could be deepened on this interesting topic, this might be a full paper per se.
>
> _It's unclear why the experiments use time on the x-axis._
>
> We display loss (or accuracy) as a function of time in order to check the overall computational burden. This is in the spirit of "frugal learning", i.e. here reducing the computational cost of Deep Learning. The main argument in that matter is however that we need a single run, there is no trial-and-error approach required to estimate a good layer width anymore, which divides the overall computational burden by the number of trials.

---

### Official Review · Reviewer_9tpv · 2022-11-01

**Confidence:** 3
**Correctness:** 3
**Technical Novelty And Significance:** 3
**Empirical Novelty And Significance:** 2
**Recommendation:** 3

**Clarity, Quality, Novelty And Reproducibility:**

The paper is clearly-written and the idea is well-motivated and, to the best of my knowledge, novel.

**Strength And Weaknesses:**

The idea is well explained and motivated, and the designed approach seems promising and novel.

The main weakness is the lack of empirical results to properly evaluate the method and test its practical implications.

To be more specific, there are previous works whose goal is to grow a network -- i.e. start from a small network and sequentially add neurons / filters / layers -- in order to save training time. These methods are typically evaluated on CIFAR-10 and ImageNet, and MNIST is too much of an easy dataset to properly evaluate these methods: the required network size to perform well on MNIST is very small. Some examples of growing methods are: MorphNet, NetAdapt, FireFly, and Net2Net.

Another concern is whether solving the quadratic problem is feasible for larger networks of different families, like CNNs.

**Summary Of The Paper:**

The paper proposes a new approach to widen layers of a neural network during training. The method is motivated by the functional gradient and looking at the desired activations for a given data input. Proof-of-concept experiments on MNIST are used to compare the method to conventional training.

**Summary Of The Review:**

See 'strengths and weaknesses' for more details.

The idea seems promising, and I suggest the authors to further investigate it through more rigorous experiments. Unfortunately it's hard to evaluate a growing method -- whose goal is to ultimately save training time by starting from a small network and growing it through training -- on a dataset as small and easy as MNIST. CIFAR-10 and ImageNet experiments are the most commonly-adopted ones when evaluating and comparing growing methods. A discussion on related works that also focus on growing networks would also be extremely valuable.

---

> ### Author Response · Authors · 2022-11-18
> **About solving the quadratic problem**
>
> We have replied to the literature and experiment concerns in our general answer to all reviewers. Regarding the specific question:
>
> _Another concern is whether solving the quadratic problem is feasible for larger networks of different families, like CNNs._
>
> Our approach is straightforwardly extendable to CNNs. Actually, during this rebuttal, we have performed experiments using CNNs on CIFAR-10, and similarly to the case of MNIST we get training curves that are comparable to / moderately above the classical training of a CNN with all neurons from the beginning (using the model size obtained at the end of our training).
>
> Regarding large networks, indeed the cost of computing a full SVD increases with the width of the network. From the beginning of this study we have thought of looking at alternative, approximate eigenvector estimation methods, such as LOBPCG (which yields a specified number of desired eigenvectors) or power iteration (which gives the first one). However, surprisingly, in practice for small/medium networks as ours, we noticed that the exact full SVD method is faster (while more accurate). Nevertheless, at some point, for large enough networks, the approximate methods will offer a better compromise between accuracy and computational time.

---

> > ### Comment · Reviewer_9tpv · 2022-12-12
> > **Response**
> >
> > Thanks for the clarifications in the response. While the idea is definitively novel and has promising aspects (e.g. accounting for redundancy), it is not possible to assess the practical implications of the method with the given experimental results. While a method might have many desirable properties in theory, an insufficient assessment of its practical performance makes it impossible to evaluate the level of contribution.
> >
> > I am keeping my score and strongly recommend the authors to expand the experiments to larger networks and more challenging datasets for future submission.

---

### Official Review · Reviewer_fbfu · 2022-11-04

**Confidence:** 2
**Correctness:** 3
**Technical Novelty And Significance:** 4
**Empirical Novelty And Significance:** 2
**Recommendation:** 3

**Clarity, Quality, Novelty And Reproducibility:**

While the approach seems to be novel and interesting I think the paper is not ready for publication and is very rushed. Experimental evaluation is also weak.

**Strength And Weaknesses:**

Strenghts:

1. Interesting Idea and compelling problem statement

Weaknesses:

1. I do not think the quality of the paper is upto ICLR standards. The writing is shabby at places with multiple typos and is difficult to follow. The way it is presented, the algorithm the authors propose is incredible hard to decipher.

2. Experimental evaluation is weak. I would expect to see results on other datasets beyond MNIST. Even for MNIST where even simple 2-layer networks get 96-97% accuracy, the plots for the classical method (standard training with fixed architecture) seems too poor for a baseline.


**Summary Of The Paper:**

The paper proposes a method to learn the optimal architecture via back propagation.

**Summary Of The Review:**

Due to concerns on presentation and experimental strengths I recommend rejection.

---

> ### Author Response · Authors · 2022-11-18
> **About results on MNIST**
>
> We have replied to the literature and experiment concerns in our general answer to all reviewers. Regarding the specific remark:
>
> _Even for MNIST where even simple 2-layer networks get 96-97% accuracy,_
>
> our point is precisely that our method, starting from one single neuron and iteratively growing, is able to achieve the same performance (96-97%) as the corresponding network (standard classic training). We are not seeking state of the art performance (which would require huge networks or extra data) but just to demonstrate that we are able to reach the standard classic training performance while starting from just one neuron. The gain with our approach is that we do not have to choose the layer widths beforehand, nor to perform multiple trainings with different widths just to find out which one yields the best accuracy/size trade-off: a single run is sufficient.
>
> For information, we have performed similar experiments using conv-nets on CIFAR-10 and we get similar results, i.e. matching classic training performance in a single run starting from a single neuron.

---

### Author Response · Authors · 2022-11-18
**General response: this is a theoretical work, the first one to avoid redundancy in neural growth, by quantifying expressivity bottlenecks**

_[TL;DR] Our work is mostly a theoretical one, and it is, to our knowledge, the first one to avoid neuron redundancy in the field of neural architecture growth._

Thank you for having pointed to relevant literature, we admit we had missed it, and it is with great enthousiasm that we see that there already exists a small community working on such topics!

We did not have the right keywords to browse through this literature as we were focusing on the conceptualization and the quantification of expressivity bottlenecks, while this literature directly targets architecture adaptation.

In this regard, we would like to emphasize that our paper is **mostly a theoretical paper**, aiming at **mathematizing the notion of expressivity bottleneck**, and at quantifying it in an easily computationable way. Indeed most theoretical work dealing with bottlenecks in neural networks either makes use of quantities such as mutual information, which are in practice very difficult to estimate (cf Information Bottlenecks), or does not take into account the Machine Learning task at hand but just studies the expressive power in general of a given architecture (which is not useful to adapt the architecture to the task). We believe we have a theoretical contribution here, which is completely absent from the neural network growth or achitecture adaptation literatures, that on the contrary build on heuristics and/or on criteria that are not justified from a functional optimization viewpoint. This yields important differences, in particular we are able to add neurons on the fly while training (without waiting for convergence) to **increase expressivity while avoiding redundancy**, whereas **the literature so far does not take into account that added neurons might be redundant with already-present neurons** (if not in a local minimum). We performed simple experiments just to show that our concepts are easily computable and usable in practice, we fully agree that they are no way sufficient to claim a new state of the art yet in neural architecture adaptation, and we actually do not claim it, this is not our main purpose in this paper. Though we believe our theoretical paper does not require extra experiments, we still try to answer the reviews in this aspect, and we are currently running experiments with VGG on CIFAR-10; unfortunately we are not sure to get the results on time. Also, this evaluation protocol, which is standard in architecture adaptation, does not make much sense to us, as it starts from an existing architecture (VGG) while the point in the long term is to actually find it.

Regarding the literature cited by reviewers:

The approaches in the architecture adaptation literature (MorphNet, AdaptNet and Net2Net) consist in exploring variations around a given (already large) architecture, relying on fully classically pre-trained networks. They most often rely on layer width exploration by trial-and-error, and re-training, which is time consuming. This is pretty different from training networks from scratch, only once, from a single neuron, and growing them, by adding the best possible neurons on the fly. Moreover, the growth moves proposed by the architecture adaptation literature are not suitable for training from a single neuron. For instance, the Net2Net growth move consists in duplicating existing layers, which is done by copying existing neurons (and adding a bit of noise). This obviously yields a lot of redundancy in the layers, instead of promoting neuron diversity in the most compact and useful manner. On the opposite, we grow networks just as much as needed.


Neural growth literature:
- Firefly is much closer to our work, as, to the opposite of the above references, it does not explore variations around a given architecture, but grows networks neuron by neuron, and this is done by estimating the best possible neuron to add, following a certain criterion. This is however done in a very different manner from ours. First, Firefly's criterion is based on loss optimization under constraints to define a small search neighborhood size, and is optimized by gradient descent. On the opposite, our criterion takes into account possible redundancy with other neurons, and we obtain the best neuron to add in close form. Moreover, we perform a line search on the amplitude of that best neuron, the consequence of which is that each addition modifies non-infinitesimally the function and accordingly improves the loss much more. More importantly, the fact that our criterion is based on expressivity bottleneck makes it usable at any time of the training, while Firefly will produce redundant neurons if the gradient descent is not finished. Consequently one can expect thinner networks with our method, and easier ways to design addition strategies (no need to wait for training convergence to a local minimum after each addition).

---

> ### Author Response · Authors · 2022-11-18
> **More on recent literature**
>
> - Concerning the work preceding Firefly: "Splitting steepest descent for growing neural architectures", one should note that it requires the computation of second-order derivatives, which is cumbersome according to the literature section of Firefly, while we require first-order derivatives only. Here again the splitting criterion does not take into account redundancy. Also, the choice of splitting neurons, i.e. making noisy copies, is an arbitrary design, while we add just plain new neurons without such constraints.
> - During this literature review we found another related paper: _"GradMax: growing neural networks using gradient information"_, published at ICLR 2022. Similarly to our work and to Firefly, it defines a criterion to optimize to find the best neuron to add. However once again, redundancy with previous neurons was forgotten in the criterion design, which leads to useless model size increase when there is no expressivity bottleneck (although it does help to decrease the loss faster).
>
> We will add a literature paragraph summing up this discussion, and emphasize that our contribution is mostly theoretical.

---

> > ### Author Response · Authors · 2022-11-19
> > **Literature added to the article**
> >
> > We added the literature to the article.

---

### Decision · Program_Chairs · 2023-01-20

**Decision:**

Reject

**Justification For Why Not Higher Score:**

Very low score.  Missing necessary empirical comparisons and literature review.

**Justification For Why Not Lower Score:**

N/A

**Metareview: Summary, Strengths And Weaknesses:**

This work develops a method for automatically growing neural networks by adding neurons "on the fly" if needed.  The reviewers all thought the approach made sense and was novel but voiced two major common concerns.  They didn't find the empirical evaluation convincing and felt that the authors didn't adequately address previous literature on adapting architectures (empirically and in literature review).  In the discussion phase, the authors acknowledged the questions regarding existing literature but argue that the work doesn't need exhaustive empirical evaluation because it is largely a theoretical contribution.  Unfortunately, the reviewers were unconvinced by this argument.

Overall, the work seems very promising but unfinished.  Placing the work in the context of the existing literature is important, not just for attribution but for the collective understanding of progress in the sub-field and ultimately to get impact for this work.  Why should others implement or build on this method if they don't know if it will work well (better than existing methods) in practice?  Therefore the recommendation is to reject, but we encourage the authors to continue developing the manuscript and hopefully the reviews will lead to a stronger submission in the future.